# Hyper-Transforming Latent Diffusion Models

**Ignacio Peis** [1 2]   **Batuhan Koyuncu** [3 4]   **Isabel Valera** [3]   **Jes Frellsen** [1 2]

## Abstract

We introduce a novel generative framework for functions by integrating Implicit Neural Representations (INRs) and Transformer-based hypernetworks into latent variable models. Unlike prior approaches that rely on MLP-based hypernetworks with scalability limitations, our method employs a Transformer-based decoder to generate INR parameters from latent variables, addressing both representation capacity and computational efficiency. Our framework extends latent diffusion models (LDMs) to INR generation by replacing standard decoders with a Transformer-based hypernetwork, which can be trained either from scratch or via *hyper-transforming*—a strategy that fine-tunes only the decoder while freezing the pre-trained latent space. This enables efficient adaptation of existing generative models to INR-based representations without requiring full retraining. We validate our approach across multiple modalities, demonstrating improved scalability, expressiveness, and generalization over existing INR-based generative models. Our findings establish a unified and flexible framework for learning structured function representations.

## 1. Introduction

Generative modelling has seen remarkable advances in recent years, with diffusion models achieving state-of-the-art performance across multiple domains, including images, videos, and 3D synthesis (Ho et al., 2020; Dhariwal & Nichol, 2021; Rombach et al., 2022). A key limitation of existing generative frameworks, however, is their reliance on structured output representations, such as pixel grids, which constrain resolution and generalisation across data modal-

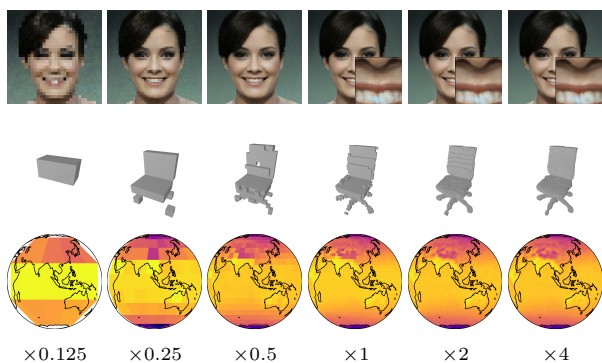

Figure 1: Samples from LDMI at multiple resolutions.

ities. In contrast, Implicit Neural Representations (INRs) have emerged as a powerful alternative that parametrises signals as continuous functions (Sitzmann et al., 2020; Mildenhall et al., 2021). By leveraging INRs, generative models can represent complex data distributions at arbitrary resolutions, yet learning expressive distributions over INR parameters remains a fundamental challenge.

A common approach to modelling distributions over INRs is to use hypernetworks (Ha et al., 2017), which generate the weights and biases of an INR conditioned on a latent code (Dupont et al., 2022b; Koyuncu et al., 2023). However, MLP-based hypernetworks suffer from scalability bottlenecks, particularly when generating high-dimensional INRs, as direct parameter regression constrains flexibility and expressiveness. More recently, Transformer-based hypernetworks have been proposed to alleviate these issues, introducing attention mechanisms to efficiently predict INR parameters (Chen & Wang, 2022; Zhmoginov et al., 2022). Nonetheless, existing approaches such as Trans-INR (Chen & Wang, 2022) remain deterministic, limiting their applicability in probabilistic frameworks.

In this work, we introduce a novel framework for INR generation, named Latent Diffusion Models of INRs (LDMI), that incorporates our proposed Hyper-Transformer Decoder (HD), a probabilistic Transformer-based decoder for learning distributions over INR parameters. Our approach combines the strengths of hypernetworks with recent advances in meta-learning for INRs while integrating into latent diffusion-based generative frameworks. Unlike previous

[1]Department of Applied Mathematics and Computer Science, Technical University of Denmark, Copenhagen, Denmark [2]Pioneer Centre for Artificial Intelligence, Copenhagen, Denmark [3]Saarland University, Saarbrücken, Germany [4]Zuse School ELIZA. Correspondence to: Ignacio Peis <ipeaz@dtu.dk>.

*Proceedings of the 42nd International Conference on Machine Learning*, Vancouver, Canada. PMLR 267, 2025. Copyright 2025 by the author(s).

works, our `HD` decoder employs a full Transformer architecture, where a Transformer Encoder processes latent variables, and a Transformer Decoder generates INR parameters via cross-attention. This design enables flexible, probabilistic generation of neural representations across diverse data modalities, surpassing the deterministic limitations of prior Transformer-based hypernetworks (Chen & Wang, 2022; Zhmoginov et al., 2022).

Our proposed framework, `LDMI`, supports two training paradigms: (i) full training, where the model is trained from scratch alongside a latent diffusion model (LDM; Rombach et al., 2022), and (ii) hyper-transforming, where a pre-trained LDM is adapted by replacing its decoder with the `HD`, allowing for efficient transfer learning without retraining the entire generative pipeline. This flexibility enables `LDMI` to scale effectively while leveraging existing pre-trained diffusion models.

**Contributions** Our key contributions can be summarized as follows:

- We introduce the `HD` decoder, a full Transformer-based probabilistic decoder for learning distributions over INR parameters.

- We integrate our method into Latent Diffusion Models (Rombach et al., 2022), with support both full training and *hyper-transforming*, enabling efficient adaptation of pre-trained models. We refer to this approach as `LDMI`.

- The `HD` decoder achieves scalability and generalisation across data modalities, overcoming the bottlenecks of MLP-based hypernetworks and extending beyond deterministic Transformer-based methods.

- We demonstrate the effectiveness of our approach on various generative tasks, showcasing its superiority in modelling high-resolution, structured data.

By integrating probabilistic INR modelling within diffusion-based generative frameworks and efficient latent-to-parameters generation, our work establishes a new direction for flexible and scalable generative modelling with unconstrained resolution. The following sections provide a detailed description of the background (Section 2), the proposed methodology (Section 3), and empirical results (Section 4).

## 2. Background and Related Work

### 2.1. Latent Diffusion Models

Latent Diffusion Models (LDM; Rombach et al., 2022) are a class of generative models that learn efficient representations by applying diffusion processes in a compressed latent space. Unlike traditional diffusion models that operate directly in high-dimensional data spaces $\boldsymbol{y} \in \mathbb{R}^D$ to approximate $p(\boldsymbol{y})$,

LDMs first encode data into a lower-dimensional latent space $\boldsymbol{z} \in \mathbb{R}^d$, where $d \ll D$, using a probabilistic encoder $\mathcal{E}_\psi(\cdot)$ to produce rich latent representations $q_\psi(\boldsymbol{z}|\boldsymbol{y})$. These are decoded via $\mathcal{D}_\lambda(\cdot)$ to accurately recover data samples by modeling the conditional distribution $p_\lambda(\boldsymbol{y}|\boldsymbol{z})$. A $\beta$-VAE (Higgins et al., 2017) with low $\beta$, a Vector-Quantized VAE (VQ-VAE; van den Oord et al., 2017), or VQGAN (Esser et al., 2021) are suitable choices for learning the structured latent space. Additionally, for image data, perceptual losses can be incorporated to further enhance generation quality (Larsen et al., 2016; Johnson et al., 2016; Hou et al., 2017; Dosovitskiy & Brox, 2016; Hou et al., 2019).

In the second stage, LDMs approximate the aggregate posterior $q_\psi(\boldsymbol{z}) = \int q_\psi(\boldsymbol{z}|\boldsymbol{y})p_{\text{data}}(\boldsymbol{y}) \, \mathrm{d}\boldsymbol{y}$ with a diffusion model $p_\theta(\boldsymbol{z})$, shifting the generative modeling task from data space to latent space and significantly reducing computational cost. While both continuous-time SDE-based approaches (Song et al., 2021b) and discrete Markov chain formulations (Ho et al., 2020; Song et al., 2021a) are viable, we follow the DDPM-based approach as adopted in the original LDM implementation (Rombach et al., 2022).

DDPMs (Ho et al., 2020) train using an objective derived from a variational lower bound on the negative log-likelihood, reparameterized for efficiency as a denoising task. This results in a simplified loss that closely resembles denoising score matching (Song et al., 2021b), and is widely used in state-of-the-art models (Saharia et al., 2022; Dhariwal & Nichol, 2021). For latent diffusion, the loss is applied to $\boldsymbol{z} \sim q_\psi(\boldsymbol{z}|\boldsymbol{y})$, yielding:

$$L_{\text{DDPM}} = \mathbb{E}_{\boldsymbol{y},\boldsymbol{z},t,\epsilon} \left[ \|\epsilon - \epsilon_\theta(\boldsymbol{z}_t, t)\|^2 \right], \quad (1)$$

where $\boldsymbol{y} \sim p_{\text{data}}(\boldsymbol{y})$, $\boldsymbol{z} \sim q_\psi(\boldsymbol{z}|\boldsymbol{y})$, $t \sim \mathcal{U}(1, T)$, and $\epsilon \sim \mathcal{N}(\boldsymbol{0}, \mathbf{I})$. This objective corresponds to predicting the added noise in a fixed forward process, effectively denoising $\boldsymbol{z}_t$ toward the original latent $\boldsymbol{z}$.

**Sampling** In DDPM, the reverse posterior density is no longer Markovian and coincides with the inference model proposed later in DDIM (Song et al., 2021a). In DDIM, it is demonstrated that faster sampling can be achieved without retraining, simply by using the posterior approximation for the estimation $\hat{\boldsymbol{z}}$, which re-defines the generative process as:

$$p_\theta(\boldsymbol{z}_{t-1}|\boldsymbol{z}_t) = \begin{cases} \mathcal{N}\left(\hat{\boldsymbol{z}}, \sigma_1^2 \boldsymbol{I}\right) & \text{if } t = 1 \\ q\left(\boldsymbol{z}_{t-1}|\boldsymbol{z}_t, \hat{\boldsymbol{z}}\right) & \text{otherwise,} \end{cases} \quad (2)$$

and uses that an estimation of $\hat{\boldsymbol{z}}$ can be computed by

$$\hat{\boldsymbol{z}} = f_\theta(\boldsymbol{z}_t, t) = \frac{1}{\sqrt{\bar{\alpha}_t}} \left( \boldsymbol{z}_t - \sqrt{1 - \bar{\alpha}_t} \cdot \epsilon_\theta(\boldsymbol{z}_t, t) \right), \quad (3)$$

where $\bar{\alpha}_t = \prod_{i=1}^{t} \alpha_i$. This formulation leads to improved efficiency using fewer steps. More details are provided in Appendix A.2.

## 2.2. Probabilistic Implicit Neural Representations

Implicit Neural Representations (INRs; Sitzmann et al., 2020) model continuous functions by mapping coordinates $\boldsymbol{x}$ to output features $\boldsymbol{y}$ via a neural network $f_\Phi(\boldsymbol{y}|\boldsymbol{x})$ (Mildenhall et al., 2021; Mescheder et al., 2019; Tancik et al., 2020). Recent generative approaches adapt this idea for flexible conditional generation at arbitrary resolutions (Dupont et al., 2022a; Koyuncu et al., 2023). Unlike models that fit $p(\boldsymbol{y})$ over structured data, INRs approximate the conditional $p(\boldsymbol{y}|\boldsymbol{x})$, allowing data sampling and uncertainty estimation at any location—especially beneficial for spatially-varying data.

Probabilistic INR generation typically involves meta-learning hidden representations that are mapped to INR parameters $\Phi$. The main challenge lies in learning these representations under uncertainty, with two goals: (i) embedding observed data into a posterior for conditional generation, and (ii) sampling from a prior for synthesis. Koyuncu et al. (2023) address this via a latent variable $\boldsymbol{z}$ and a variational framework combining a flow-based prior $p_\theta(\boldsymbol{z})$ with an encoder $q_\psi(\boldsymbol{z}|\boldsymbol{x}, \boldsymbol{y})$, using an MLP-based hypernetwork to map $\boldsymbol{z}$ to $\Phi$ (Ha et al., 2017). However, scaling to high-resolution images is limited by the complexity of the INR parameter space. Similarly, Dupont et al. (2022b) use adversarially trained hypernetworks, which improve stability but lack conditional generation and face the same scalability bottlenecks.

Two recent strategies aim to address these issues. Functa (Dupont et al., 2022a; Bauer et al., 2023) learns neural fields per datapoint via optimization of modulation codes (referred to as *functas*), then trains a generative model on the resulting *functaset*. Alternatively, Chen et al. (2024) leverage LDMs by pretraining an autoencoder (e.g., $\beta$-VAE or VQ-VAE) with an image renderer incorporated to the decoder, and applying a diffusion model on its latent space.

## 2.3. Hyper-Transformers

Hypernetworks (Ha et al., 2017) map latent representations $\boldsymbol{z}$ to INR parameters $\Phi = g_\phi(\boldsymbol{z})$, where $g_\phi$ is a learned generator and $f_\Phi(\boldsymbol{y}|\boldsymbol{x})$ defines the INR. While the hypernetwork is shared across the dataset, the INR is instance-specific, generated dynamically by the hypernetwork. This design, however, struggles with large INRs, as generating full parameter vectors introduces capacity and optimization bottlenecks. To overcome this, recent work has proposed Hyper-Transformers (Chen & Wang, 2022; Zhmoginov et al., 2022), which use attention to generate INR parameters in modular chunks. This design avoids flattening and allows scalable, targeted weight generation.

Building on this paradigm, recent work, (HyperDream-Booth; Ruiz et al., 2024) extends this idea by generating

LoRA adapters for personalizing diffusion models from few examples. While their focus is efficient adaptation, our framework (LDMI) builds a full generative model over INR parameter space. Our Hyper-Transformer Decoder enables resolution-agnostic generation across modalities (images, 3D fields, climate data), representing a new direction in function-level generative modeling.

Although prior methods succeed in deterministic reconstruction, they do not support probabilistic generation or serve as decoders for latent variable models. Our approach addresses this by using full Transformer architectures conditioned on tensor-shaped latent variables.

## 2.4. Function-valued Stochastic Processes

Alternative approaches like Neural Diffusion Processes (NDPs) (Dutordoir et al., 2023) and Simformer (Gloeckler et al., 2024) define distributions over function values, not over network parameters. NDPs denoise function values directly using permutation-invariant attention, mimicking Neural Processes, while Simformer learns a joint diffusion model over data and simulator parameters. In contrast, our method directly generates INR parameters via a hyper-transformer, offering continuous function generation with resolution independence.

## 3. Methodology

We introduce Latent Diffusion Models of Implicit Neural Representations (LDMI), a novel family of generative models operating in function space. While previous probabilistic INR approaches have explored latent variable models for INRs (Koyuncu et al., 2023; Dupont et al., 2022b) and two-stage training schemes incorporating diffusion-based priors in the second stage (Park et al., 2023; Dupont et al., 2022a; Bauer et al., 2023), these methods face key limitations. Notably, MLP-based hypernetworks suffer from capacity bottlenecks, while existing frameworks lack a unified and compact approach for handling diverse data modalities.

To address these challenges, we propose the Hyper-Transformer Decoder (HD), which extends the flexibility of hypernetworks with the scalability of Transformer-based architectures. Unlike prior deterministic frameworks (Chen & Wang, 2022), HD introduces a probabilistic formulation, enabling uncertainty modelling and improved generative capacity for INRs.

### 3.1. Notation

We define $\mathcal{F} = \{f : \mathcal{X} \to \mathcal{Y}\}$ as the space of continuous signals, where $\mathcal{X}$ represents the domain of coordinates, and $\mathcal{Y}$ the codomain of signal values, or features. Let $\mathcal{D}$ be a dataset consisting of $N$ pairs $\mathcal{D} \doteq \{(\boldsymbol{X}_n, \boldsymbol{Y}_n)\}_{n=1}^N$. Here, $\boldsymbol{X}_n$ and $\boldsymbol{Y}_n$ conform a signal as a collection of $D_n$

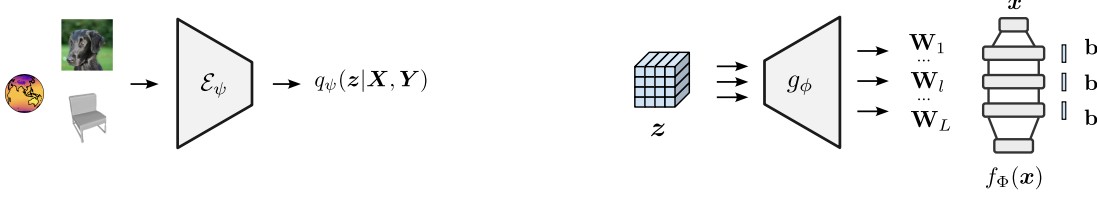

(a) Encoder.  (b) INR Generation.

Figure 2: Overview of the `I-VAE`. (a) The inference model, trained in the first stage, where an encoder maps data into the variational parameters of the approximate posterior; (b) The full decoder: the `HD` module transforms the latent variable into the weights and biases of an INR, enabling continuous signal representation.

coordinates $\boldsymbol{X}_n \doteq \{\boldsymbol{x}_i^{(n)}\}_{i=1}^{D_n}$, $\boldsymbol{x}_i^{(n)} \in \mathcal{X}$ and signal values $\boldsymbol{Y}_n \doteq \{\boldsymbol{y}_i^{(n)}\}_{i=1}^{D_n}$, $\boldsymbol{y}_i^{(n)} \in \mathcal{Y}$.

### 3.2. Generative Model

We aim to model the stochastic process that generates continuous signals $(\boldsymbol{X}, \boldsymbol{Y})$ using neural networks $p(\boldsymbol{y}|\boldsymbol{x}; \Phi) \equiv p_\Phi(\boldsymbol{y}|\boldsymbol{x})$, known as Implicit Neural Representations (INRs). Unlike discrete representations such as pixel grids, INRs provide compact and differentiable function approximations, making them well-suited for a wide range of data modalities, including images (Chen et al., 2021), 3D shapes (Mescheder et al., 2019), and audio (Sitzmann et al., 2020), while inherently supporting unconstrained resolution.

**Implicit Neural Representation**  Previous works have demonstrated that even simple MLP-based architectures exhibit remarkable flexibility, accurately approximating complex signals. In this work, we define our INR as an MLP with $L$ hidden layers:

$$\begin{aligned}
\boldsymbol{h}_0 &= \gamma(\boldsymbol{x}), \\
\boldsymbol{h}_l &= \sigma(\mathbf{W}_l \boldsymbol{h}_{l-1} + \mathbf{b}_l), \quad l = 1, \dots, L, \\
f_\Phi(\boldsymbol{x}) &= \boldsymbol{h}_L = \mathbf{W}_L \boldsymbol{h}_{L-1} + \mathbf{b}_L.
\end{aligned} \quad (4)$$

where $\gamma$ denotes optional coordinate encoding, and $\sigma$ is a non-linearity. The INR parameters are given by $\Phi = (\mathbf{W}_l, \mathbf{b}_l)_{l=1}^L$. The network parametrise a likelihood distribution, $\lambda$, over the target signal, that is,

$$p_\Phi(\boldsymbol{y}|\boldsymbol{x}) = \lambda(\boldsymbol{y}; f_\Phi(\boldsymbol{x})). \quad (5)$$

**Challenges in Modeling INR Parameter Distributions** Learning a generative model over the INR parameter space $\Phi$ is highly non-trivial due to two fundamental challenges. First, small perturbations in parameter space can result in drastic variations in the data space, making direct modelling difficult. Second, the dimensionality of the flattened parameter vector $\Phi$ scales poorly with the INR's width and depth, leading to significant computational and optimization challenges.

A common approach in prior works (Koyuncu et al., 2023; Dupont et al., 2022b;a) is to model $\Phi$ implicitly using an auxiliary neural network known as a *hypernetwork* (Ha et al., 2017), typically MLP-based, which modulates the INR parameters based on a lower-dimensional latent representation:

$$\Phi = g_\phi(\boldsymbol{z}), \quad (6)$$

where $\boldsymbol{z} \in \mathbb{R}^{H_z \times W_z \times d_z}$ is a tensor-shaped latent code with spatial dimensions $H_z \times W_z$ and channel dimension $d_z$. This latent space serves as a compressed representation of the continuous signal. However, MLP-based hypernetworks introduce a fundamental bottleneck: the final layer must output all parameters of the target INR, which leads to scalability issues when modulating high-capacity INRs. As a result, prior works employing MLP-based hypernetworks typically use small INR architectures, often restricted to three-layer MLPs (Koyuncu et al., 2023; Dupont et al., 2022b). Additionally, these works rely on standard MLPs with ReLU or similar activations and often preprocess coordinates using Random Fourier Features (RFF) (Tancik et al., 2020) to capture high-frequency details. However, RFF-based embeddings struggle to generalize to unseen coordinates without careful validation. In contrast, SIREN (Sitzmann et al., 2020) inherently captures the full frequency spectrum by incorporating sinusoidal activations, making it more effective for super-resolution tasks.

However, using SIREN as the INR module introduces further challenges. As commented by their authors, optimizing SIRENs with not carefully chosen uniformly distributed weights yields poor performance both in accuracy and in convergence speed. This issue worsens when the weights are not optimized, but generated by a hypernetwork.

To overcome these limitations, we introduce the Hyper-Transformer Decoder (`HD`), a Transformer-based hypernetwork designed to scale effectively while preserving the flexibility of larger INRs and ensuring proper modulation of SIREN weights.

#### 3.2.1. HYPER-TRANSFORMER DECODER

The Hyper-Transformer Decoder (`HD`) is a central component of our autoencoding framework for generating INRs, referred to as `I-VAE` and illustrated in Figure 2. Its inter-

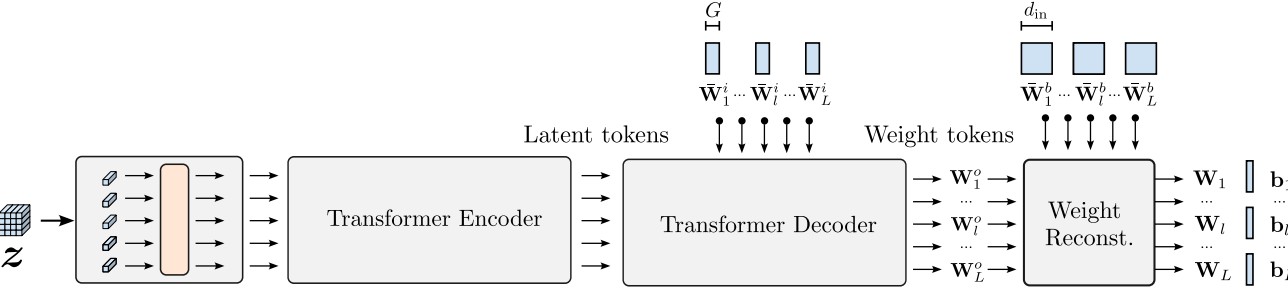

Figure 3: Diagram of the Hyper-Transformer Decoder (HD). The latent variable $z$ is tokenized and processed by a Transformer Encoder. A Transformer Decoder, initialized with learnable grouped weights $\bar{\mathbf{W}}_l^i$, cross-attends to the latent tokens to generate the set of grouped weights $\mathbf{W}_l^o$. The full weight matrices $\mathbf{W}_l$ are then reconstructed by combining the grouped weights with learnable template weights $\bar{\mathbf{W}}_l^b$. Biases $\mathbf{b}_l$ are learned as global parameters.

nal architecture is detailed in Figure 3. Mathematically, it is defined as a function of the latent tensor that produces the parameters $\Phi = g_\phi(z)$ of the INR. We elaborate on previous work that proposed Vision Transformers (Dosovitskiy et al., 2021) for meta-learning of INR parameters from images (Chen & Wang, 2022), to design an efficient full Transformer architecture (Vaswani et al., 2017) that processes latent variables into INR parameters.

**Tokenizer** The HD decoder begins by splitting tensor-shaped latent variables into $N$ patches $z_p \in \mathbb{R}^{P^2 \times d_z}$ of fixed size $(P, P)$, where $N = H_z \cdot W_z / P^2$. Each patch is then flattened and projected into a lower-dimensional embedding space using a shared linear transformation. The embedding for each patch serves as an input token for the Transformer.

**Transformer Encoder** Following tokenization, the first half of the HD decoder, a Transformer Encoder, applies multi-head self-attention mechanism, repeated for several layers, outputting tokenized embeddings that we referred to as *latent tokens*.

**Transformer Decoder** Unlike prior work (Chen & Wang, 2022) that only employs Transformer Encoders, we introduce a Transformer Decoder that cross-attends to latent tokens to generate output tokens, which are then mapped to the column weights of the INR weight matrices. The input to the Transformer Decoder consists of a globally shared, learnable set of initial weight tokens, denoted as $\bar{\mathbf{W}}^i$, which serve as queries. The decoder processes these queries via cross-attention with the latent tokens, which act as keys and values to obtain the output set of column weights, $\mathbf{W}^o$.

**Weight Grouping** Following Chen et al. (2021), we adopt a similar weight grouping strategy to balance precision and computational efficiency in INR parameter generation. However, our approach differs in the reconstruction strategy applied to the Transformer-generated weights.

Let $\mathbf{W} \in \mathbb{R}^{d_{out} \times d_{in}}$ represent a weight matrix of an INR layer (omitting the layer index $l$ for simplicity), where each column is denoted as $w_c$ for $c \in \{1, \ldots, d_{in}\}$. Directly mapping a Transformer token to every column is computationally expensive, as it would require processing long sequences. Instead, we define $G$ groups per weight matrix, where each group represents $k = \frac{d_{in}}{G}$ columns (assuming divisibility). This results in a Transformer working with grouped weight matrices $\bar{\mathbf{W}}^i, \mathbf{W}^o \in \mathbb{R}^{d_{out} \times G}$, since $\bar{\mathbf{W}}^i$ (input tokens) and $\mathbf{W}^o$ (output tokens) have the same dimensions. The full weight matrix is reconstructed as:

$$w_c = \mathcal{R}(w^o_{\lceil c/k \rceil}, \bar{w}^b_c) \tag{7}$$

where $\mathcal{R}$ is the *reconstruction* operator, $w_c$ denotes the $c$th column of $\mathbf{W}$, and $\bar{w}^b_c$ is the corresponding column of $\bar{\mathbf{W}}^b \in \mathbb{R}^{d_{out} \times d_{in}}$—a set of learnable base parameters that serve as a weight template.

In Trans-INR (Chen & Wang, 2022), the Transformer-based hypernetwork generates normalized weights using the following reconstruction method:

$$\mathcal{R}^{(norm)}\left(w^o_{\lceil c/k \rceil}, \bar{w}^b_c\right) = \frac{w^o_{\lceil c/k \rceil} \odot \bar{w}^b_c}{\left\| w^o_{\lceil c/k \rceil} \odot \bar{w}^b_c \right\|}. \tag{8}$$

However, we found this approach unsuitable for generating weights in INRs with periodic activations. We hypothesize that the normalization in Equation (8) causes the resulting weights and their gradients to vanish when $\|w^o_{\lceil c/k \rceil}\| \approx 0$, leading to training instability.

To address this, we introduce a new reconstruction operator that removes the normalization constraint, yielding significantly more stable training, particularly for INRs with periodic activations:

$$\mathcal{R}^{(scale)}\left(w^o_{\lceil c/k \rceil}, \bar{w}^b_c\right) = (1 + w^o_{\lceil c/k \rceil}) \odot \bar{w}^b_c. \tag{9}$$

This formulation ensures that when $\|w^o_{\lceil c/k \rceil}\| \approx 0$, the base weights remain unchanged, preventing instability in the generated INR parameters.

It is essential to distinguish between the two sets of globally shared learnable parameters. The sequence $\bar{\mathbf{W}}^{\mathrm{i}}_{1:L}$ (of length $G \times L$) serves as the initialization for the grouped weight tokens and is provided as input to the Transformer Decoder. In contrast, the sequence $\bar{\mathbf{W}}^{\mathrm{b}}_{1:L}$ (of length $d_{in} \times L$) acts as a global reference for the full set of INR weights and is used exclusively for the reconstruction in Equation (9).

This weight grouping mechanism allows the `HD` decoder to dynamically adjust the trade-off between precision and efficiency by tuning $G$. As a result, our method scales effectively across different model sizes while preserving the benefits of Transformer-based hypernetwork parameterization.

### 3.3. Variational Inference

Given our implicit modelling of INR parameters, the objective is to learn a structured and compact probabilistic latent space $p_\theta(z)$ that enables meaningful posterior inference via Bayesian principles. Specifically, we seek to approximate the true posterior $p(z|X, Y)$, facilitating accurate and flexible inference. To achieve this, we extend the Variational Autoencoder (VAE) framework (Kingma & Welling, 2013) to the efficient generation of INRs, which we refer to as `I-VAE` (depicted in Figure 2).

The complexity of the likelihood function makes the true posterior $p(z|X, Y)$ intractable. To address this, VAEs introduce an encoder network that approximates the posterior through a learned variational distribution. Our encoder, denoted as $\mathcal{E}_\psi(X, Y)$, processes both coordinate inputs $X$ and signal values $Y$, outputting the parameters of a Gaussian approximation:

$$q_\psi(z|X, Y) = \mathcal{N}\left(z;\ \mathcal{E}_\psi(X, Y)\right). \qquad (10)$$

This formulation enables training via amortized variational inference, optimising a lower bound on the log-marginal likelihood $\log p(Y|X)$, known as the Evidence Lower Bound (ELBO):

$$\begin{aligned}
\mathcal{L}_{\mathrm{VAE}}(\phi, \psi) = \ &\mathbb{E}_{q_\psi(z|X,Y)}\left[\log p_\Phi(Y|X)\right] \\
&- \beta \cdot D_{\mathrm{KL}}\left(q_\psi(z|X, Y) \,\|\, p(z)\right),
\end{aligned} \qquad (11)$$

where we omit the explicit dependence of $\Phi$ on $\phi$ and $z$ for clarity. The hyperparameter $\beta$, introduced by Higgins et al. (2017), controls the trade-off between reconstruction fidelity and latent space regularization, regulating the amount of information compression in the latent space. The case $\beta = 1$ corresponds to the standard definition of the ELBO.

During the first stage of training, following LDMs (Rombach et al., 2022), we impose a simplistic standard Gaussian prior $p(z) = \mathcal{N}(0, I)$, and we set a low $\beta$ value to encourage high reconstruction accuracy while promoting a structured latent space that preserves local continuity. This

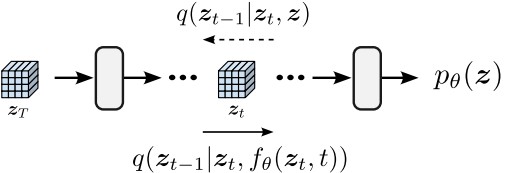

Figure 4: Schematic of the DDIM-based latent diffusion.

choice facilitates smooth interpolations and improves the quality of inferred representations.

In addition, following Rombach et al. (2022), for the case of images, we incorporate perception losses (Zhang et al., 2018), and patch-based (Isola et al., 2017) adversarial objectives (Dosovitskiy & Brox, 2016; Esser et al., 2021; Yu et al., 2022).

However, while this training strategy ensures effective inference, direct generation remains poor due to the discrepancy between the expressive encoder distribution and the simplistic Gaussian prior. To address this mismatch, we introduce a second training stage, where a diffusion-based model is fitted to the marginal posterior distribution, enhancing the generative capacity of the learned latent space.

### 3.4. Latent Diffusion

In this stage, we leverage the pre-trained `I-VAE` and the highly structured latent space encoding INRs to fit a DDPM (Ho et al., 2020) to the aggregate posterior:

$$q_\psi(z) = \mathbb{E}_{p_{\mathrm{data}}(X, Y)}\left[q_\psi(z|X, Y)\right]. \qquad (12)$$

In other words, we fit a learnable prior over the latent space to approximate the structured posterior induced by the encoder. Specifically, we minimize the Kullback-Leibler divergence

$$D_{\mathrm{KL}}\left(q_\psi(z) \,\|\, p_\theta(z)\right), \qquad (13)$$

where $p_\theta(z)$ represents the learned diffusion-based prior over the latent space. We instantiate the following variation of the Denoising Score Matching (DSM; Vincent, 2011) objective presented in Equation (1):

$$\mathcal{L}_{\mathrm{DDPM}} = \mathbb{E}_{X,Y,z,\epsilon,t}\left[\lambda(t)\|\epsilon - \epsilon_\theta(z_t, t)\|^2\right], \qquad (14)$$

which is approximated via Monte Carlo sampling. Specifically, we draw $(X, Y) \sim p_{\mathrm{data}}(X, Y)$, $z \sim q_\psi(z|X, Y)$, $\epsilon \sim \mathcal{N}(0, I)$, and $t \sim U(1, T)$, where $T$ denotes the number of diffusion steps. The weighting function $\lambda(t)$ modulates the training objective; in our case, we set $\lambda(t) = 1$, corresponding to an unweighted variational bound that enhances sample quality. The complete training procedure for `LDMI` is outlined in Appendix B.

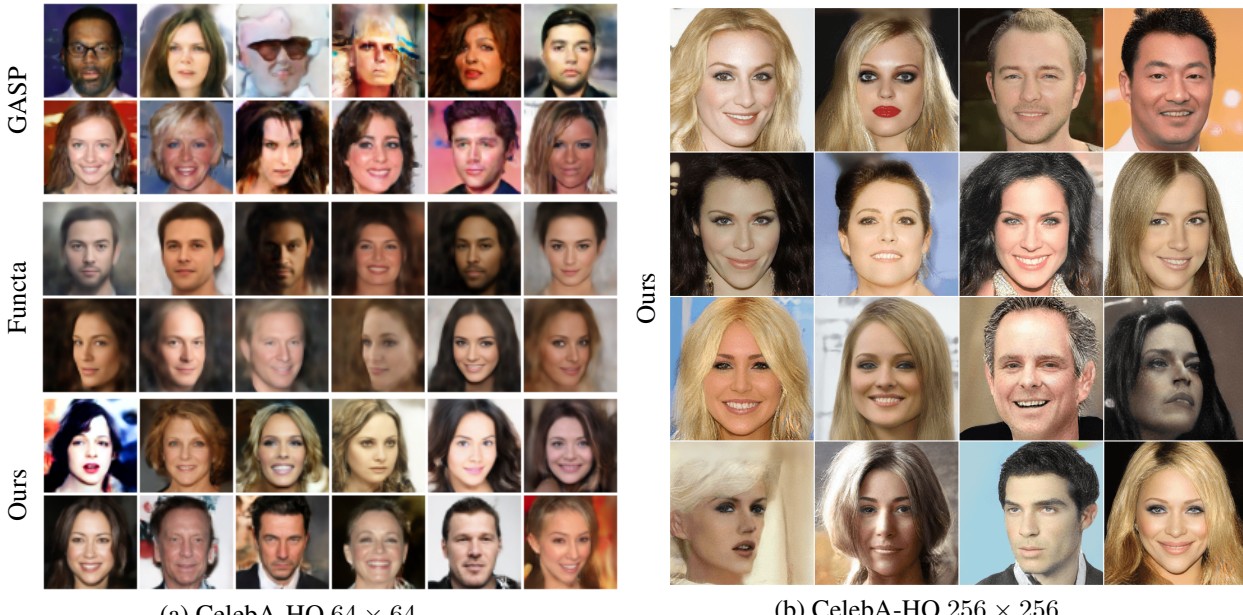

(a) CelebA-HQ $64 \times 64$        (b) CelebA-HQ $256 \times 256$

Figure 5: Uncurated samples from GASP, Functa (diffusion-based) and our `LDMI` trained with CelebA-HQ at $64\times64$ resolution (a). In (b), our model was trained on CelebA-HQ at $256\times256$ using the hyper-transforming approach.

## 3.5. Hyper-Transforming Latent Diffusion Models

As an alternative to full training, we introduce a highly efficient approach for training `LDMI` when a pre-trained LDM for discretized data is available. In this setting, we eliminate the need for two-stage training by leveraging the structured latent space learned by the pre-trained LDM. Specifically, we freeze the VAE encoder and the LDM and train only the `HD` decoder to maximize the log-likelihood of the decoded outputs, using the objective

$$\mathcal{L}_{\text{HT}}(\phi) = \mathbb{E}_{q_\psi(\boldsymbol{z}|\boldsymbol{X},\boldsymbol{Y})} \left[ \log p_\Phi(\boldsymbol{Y}|\boldsymbol{X}) \right], \quad (15)$$

optionally combined with perceptual and adversarial objectives. The `HD` decoder is sufficiently expressive to extract meaningful information from the pre-trained latent space.

We refer to this strategy as *hyper-transforming*, as it enables efficient adaptation of a standard LDM to an INR-based framework without retraining the latent encoder or diffusion prior. Notably, this approach is compatible with LDMs based on VAEs, VQVAEs, or VQGANs—since all that is required is access to a pre-trained encoder and diffusion model. The full procedure is outlined in Algorithm 1.

## 4. Experiments

### 4.1. Training Setup

Depending on the nature of the data, we utilize different architectures for the encoder. For image and climate data, we employ ResNets (He et al., 2016), while for 3D occupancy data, we employ 3D-convolutional networks.

Our evaluations[1] span multiple domains: (1) natural image datasets, including CelebA-HQ (Liu et al., 2015) at several resolutions and ImageNet (Russakovsky et al., 2015); (2) 3D objects, specifically the Chairs subclass from the ShapeNet repository (Chang et al., 2015), which provides approximately 6,778 chair models for 3D reconstruction and shape analysis; and (3) polar climate data, using the ERA5 temperature dataset (Hersbach et al., 2019) to analyze global climate dynamics. In our hyper-transforming setup, we leverage publicly available pre-trained Latent Diffusion Models from Rombach et al. (2022), specifically the LDM-VQ-4 variants trained on CelebA-HQ at $256 \times 256$ resolution and ImageNet, respectively.

### 4.2. Generation

Figure 1 shows samples generated by `LDMI` across diverse data modalities and resolutions. To obtain these, we sample from the latent diffusion model, decode the latents into INRs using our `HD` decoder, and evaluate the resulting functions on different coordinate grids. Notably, CelebA-HQ at $(256 \times 256)$ is a challenging dataset not addressed by prior baselines.

In Figure 5, we provide samples from `LDMI` trained on CelebA at $64\times64$ and $256\times256$, demonstrating its ability to generate high-quality and diverse images across different scales, and the qualitative superiority against the baselines. Additionally, in Table 1, we report FID scores of our sam-

---

[1]The code for reproducing our experiments can be found at `https://github.com/ipeis/LDMI`.

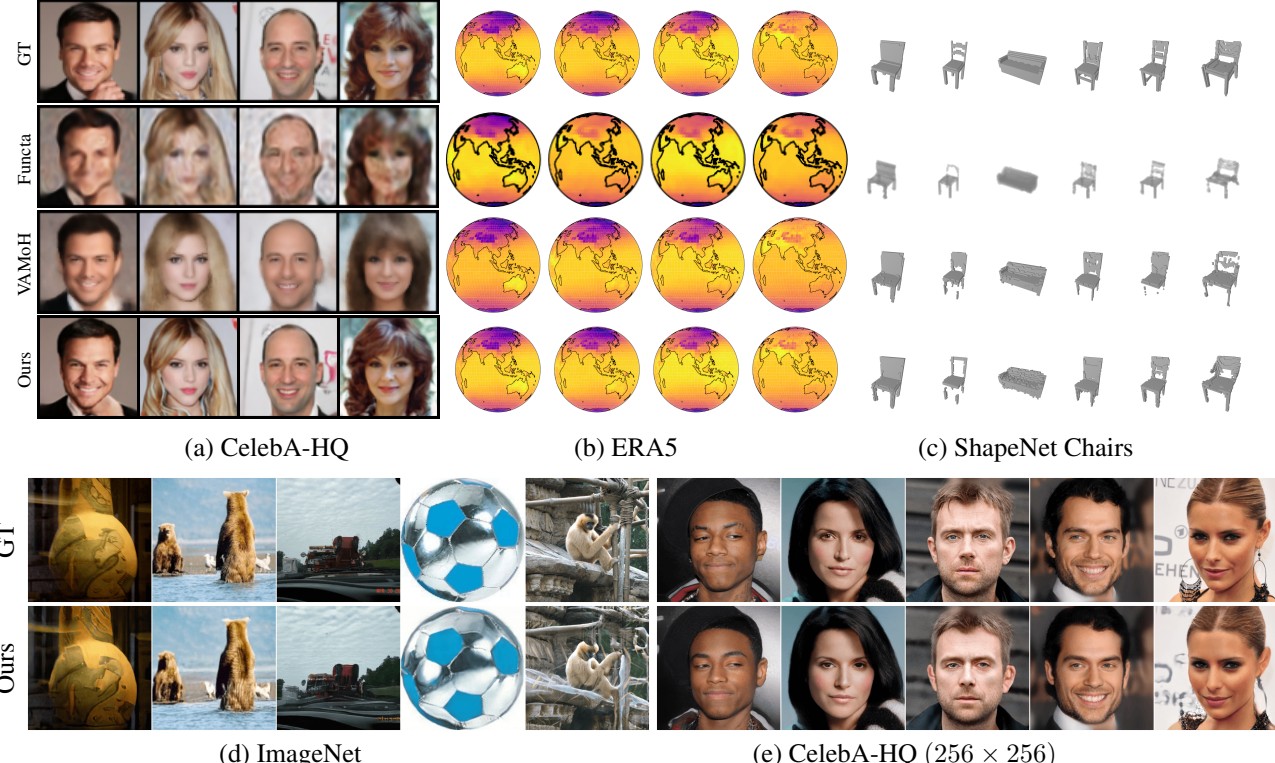

(a) CelebA-HQ      (b) ERA5      (c) ShapeNet Chairs

(d) ImageNet      (e) CelebA-HQ $(256 \times 256)$

Figure 6: Reconstructions from Functa (diffusion-based), VAMoH, and our `LDMI` compared to ground truth (GT) across three datasets (a)–(c). In (d) and (e) `LDMI` was trained by hyper-transforming pre-trained LDMs.

| Model | PSNR (dB) ↑ | FID ↓ | HN Params ↓ |
|---|---|---|---|
| **CelebA-HQ** $(64 \times 64)$ | | | |
| GASP (Dupont et al., 2022b) | - | **7.42** | 25.7M |
| Functa (Dupont et al., 2022a) | ≤ **30.7** | 40.40 | - |
| VAMoH (Koyuncu et al., 2023) | 23.17 | 66.27 | 25.7M |
| LDMI | 24.80 | 18.06 | **8.06M** |
| **ImageNet** $(256 \times 256)$ | | | |
| Spatial Functa (Bauer et al., 2023) | ≤ 38.4 | ≤ 8.5 | - |
| LDMI | 20.69 | **6.94** | 102.78M |

Table 1: Metrics on CelebA-HQ and ImageNet.

ples for CelebA and ImageNet, highlighting the model's performance in terms of image quality and reconstruction accuracy. Importantly, while GASP achieves a lower FID score on CelebA-HQ, it cannot perform reconstructions due to its adversarial design. In contrast, our `LDMI` supports both high-quality sampling and accurate reconstructions, as demonstrated in the following experiment.

### 4.3. Reconstruction

Figure 6 presents a qualitative comparison between original images from the test split and their reconstructions by our model. As shown in the figure, our model successfully preserves fine details and global structures across different images. On ImageNet, our model demonstrates robust re-

construction performance across diverse object categories and scenes, including textures, object boundaries and colour distributions.

Quantitative evaluations in Table 1 indicate competitive performance compared to existing INR-based generative models. For CelebA-HQ $64 \times 64$, we achieve a PSNR value of 24.80 dB, which compares competitively to the previous methods. The high PSNR score suggests there is minimal information loss during the encoding-decoding process while maintaining perceptual quality. It is important to note several key distinctions when interpreting the comparative results in this Table. While Functa exhibits higher PSNR values, this advantage stems from its test-time optimization procedure—fitting a separate modulation vector per test image using ground truth information—rather than the amortized inference approach employed by our model. This fundamental methodological difference undermines a direct comparison, though we include these results for completeness.

As for generation, our model enables seamless reconstruction at arbitrary resolutions. Figures 9 and 10 show results from `LDMI` trained on CelebA-HQ at $(256 \times 256)$ and $(64 \times 64)$, respectively. To achieve this, we encode the observations, sample from the posterior distribution, and transform these latent samples into INRs using our HD de-

coder. The resulting INRs are then evaluated on grids of varying resolutions to produce reconstructions ranging from $\times 1/8$ to $\times 4$. Importantly, such flexible resolution control is not supported by methods like GASP, which cannot perform reconstructions due to their adversarial design.

To further validate our claims regarding cross-modality generalization, we evaluate LDMI on two additional, diverse datasets: ShapeNet Chairs and ERA5 climate data. As shown in Table 2, while Functa achieves better performance on Chairs—possibly due to being optimization-based—LDMI attains higher reconstruction quality than VA-MoH (inference-based) across both datasets, outperforming it by 0.5% on Chairs and 5.6 dB on ERA5. The substantial performance gain on the ERA5 climate dataset highlights LDMI's exceptional capability in representing complex, continuous spatio-temporal signals beyond standard visual data, further supporting our argument for a truly general-purpose INR-based generative framework.

| Model | Chairs (%) ↑ | ERA5 (dB) ↑ |
|---|---|---|
| Functa (Dupont et al., 2022a) | **99.51** | 34.9 |
| VAMoH (Koyuncu et al., 2023) | 96.75 | 39.0 |
| LDMI | 97.25 | **44.6** |

Table 2: Reconstruction quality (accuracy in % and PSNR in dB) on ShapeNet Chairs and ERA5 climate data, demonstrating LDMI's strong generalization capabilities across modalities. Note that GASP is omitted as it is not applicable to INR reconstruction tasks.

The dual capability of high-quality generation and accurate reconstruction positions LDMI as a versatile foundation for various downstream tasks requiring both generative and reconstructive abilities, a unique advantage over specialized alternatives optimized for only one of these objectives.

### 4.4. Hyper-Transforming

On ImageNet and CelebA-HQ at $256 \times 256$, our LDMI was trained using the *hyper-transforming* strategy. As discussed in Section 3.5, instead of training the model from scratch, we reuse the encoder and diffusion model from a pre-trained LDM, training only the Hyper-Transformer Decoder. This approach leverages both the structured latent space learned by the autoencoder and the diffusion model already fitted to it. Training then focuses solely on transforming latent codes into the function space of INR parameters, enabling efficient specialization without retraining the generative backbone.

Our method is agnostic to the latent encoder—VAE, VQ-VAE, or VQGAN—highlighting the flexibility of the hyper-transforming framework. These results demonstrate its scalability for INR-based synthesis and adaptation of pre-trained LDMs to resolution-agnostic generation.

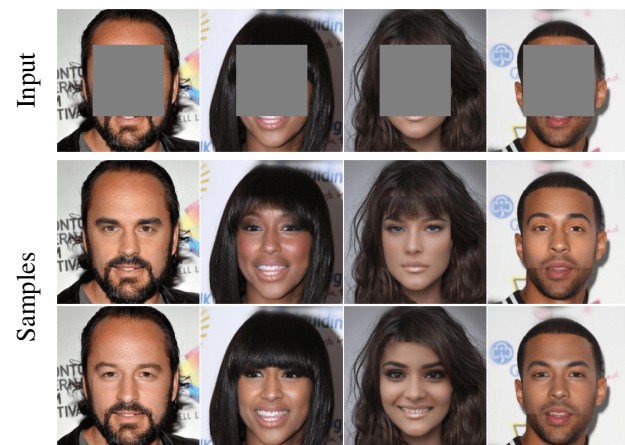

Figure 7: Conditional inpainting results on CelebA-HQ at $256 \times 256$. The second and third rows present two conditional samples generated by our LDMI for the missing regions at the centre.

### 4.5. Data Completion

We evaluate our model on structured completion tasks, where parts of the input are masked with predefined patterns. For each case, we show the masked input and two samples drawn from the posterior. As shown in Figure 7, the model uses visible context to produce realistic, coherent outputs. On CelebA-HQ $256 \times 256$, it reconstructs missing features—such as eyes, mouth, or hairline—while preserving consistency with the unmasked regions. These results highlight its ability to generalize from partial observations.

## 5. Conclusion

We introduced Latent Diffusion Models of Implicit Neural Representations (LDMI), a framework that combines the expressiveness of INRs with the generative power of Latent Diffusion Models. Our hyper-transforming approach allows efficient adaptation of pre-trained LDMs to INR-based generation without retraining the full model. Experiments on CelebA-HQ, ImageNet, ShapeNet Chairs, and ERA5 demonstrate strong performance in both quantitative and qualitative evaluations.

Importantly, LDMI overcomes limitations of prior hypernetworks through a modular, probabilistic Transformer decoder that supports scalable, expressive INR generation. While MLP-based methods struggle with dense parameter outputs and Transformer-based ones are typically deterministic, our design enables probabilistic modelling and efficient synthesis. Integrating INRs with latent diffusion yields a flexible, resolution-agnostic framework—advancing generative tasks requiring structural consistency, fidelity, and scalability.

## Acknowledgements

This research was supported by the Villum Foundation through the Synergy project number 50091, by the Novo Nordisk Foundation through the Center for Basic Machine Learning Research in Life Science (MLLS, grant NNF20OC0062606) and by the European Union (ERC-2021-STG, SAML, 101040177).

Ignacio Peis acknowledges support by Danish Data Science Academy, which is funded by the Novo Nordisk Foundation (NNF21SA0069429). Batuhan Koyuncu is supported by the Konrad Zuse School of Excellence in Learning and Intelligent Systems (ELIZA) through the DAAD programme Konrad Zuse Schools of Excellence in Artificial Intelligence, sponsored by the Federal Ministry of Education. The views and opinions expressed are those of the author(s) only and do not necessarily reflect those of the European Union or the European Research Council Executive Agency. Neither the European Union nor the granting authority can be held responsible.

## Impact Statement

This paper presents work whose goal is to advance the field of Machine Learning. There are many potential societal consequences of our work, none which we feel must be specifically highlighted here.

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

## A. Background Extension

### A.1. Autoencoding-based Models

When opting by $\beta$-VAE (Higgins et al., 2017) for the first stage of LDMs (Rombach et al., 2022), a weak regularization encourages accurate likelihoods $p_\lambda(\boldsymbol{y}|\boldsymbol{z})$ parameterized by the decoder $D_\lambda(\cdot)$ while preserving latent expressiveness. This is achieved by assuming $p(\boldsymbol{z}) = \mathcal{N}(\boldsymbol{0}, \boldsymbol{I})$ and maximizing the Evidence Lower Bound (ELBO):

$$\mathcal{L}_{\text{VAE}} = \mathbb{E}_{q_\psi(\boldsymbol{z}|\boldsymbol{y})}\left[\log p_\lambda(\boldsymbol{y}|\boldsymbol{z})\right] - \beta \cdot D_{\text{KL}}\left(q_\psi(\boldsymbol{z}|\boldsymbol{y}) \,\|\, p(\boldsymbol{z})\right) \tag{16}$$

and setting $\beta$ is set to a small value to encourage accurate reconstructions.

In contrast, VQVAE replaces the continuous latent representation with a discrete codebook. During training, the encoder output is quantized using the closest embedding from the codebook, enabling a more structured representation. The objective becomes:

$$\mathcal{L}_{\text{VQVAE}} = \|\boldsymbol{y} - D_\lambda(E_\phi(\boldsymbol{y}))\|^2 + \|\text{sg}[E_\phi(\boldsymbol{y})] - \boldsymbol{z}\|^2 + \|\text{sg}[\boldsymbol{z}] - E_\phi(\boldsymbol{y})\|^2 \tag{17}$$

where $\text{sg}[\cdot]$ indicates a stop-gradient operation that prevents the encoder from directly updating the codebook embeddings.

### A.2. Diffusion Models

In DDPM, the forward diffusion process incrementally corrupts the observed variable $\boldsymbol{z}_0$ using a Markovian noise process with latent variables $\boldsymbol{z}_{1:T}$

$$q(\boldsymbol{z}_{1:T}|\boldsymbol{z}_0) := \prod_{t=1}^{T} q(\boldsymbol{z}_t|\boldsymbol{z}_{t-1}), \tag{18}$$

where $q(\boldsymbol{z}_t|\boldsymbol{z}_{t-1}) := \mathcal{N}(\boldsymbol{z}_t; \sqrt{1-\beta_t}\boldsymbol{z}_{t-1}, \beta_t\mathbf{I})$, and $\beta_t$ is a noise schedule controlling the variance at each time step.

The reverse generative process is defined by the following Markov chain

$$p_\theta(\boldsymbol{z}_{0:T}) := p(\boldsymbol{z}_T)\prod_{t=1}^{T} p_\theta(\boldsymbol{z}_{t-1}|\boldsymbol{z}_t) \tag{19}$$

where $p_\theta(\boldsymbol{z}_{t-1}|\boldsymbol{z}_t) := \mathcal{N}\left(\mu_\theta(\boldsymbol{z}_t, t), \boldsymbol{\sigma}_t^2\right)$ is defined as a Gaussian whose mean

$$\mu_\theta(\boldsymbol{z}_t, t) = \frac{1}{\sqrt{\alpha_t}}\left(\boldsymbol{z}_t - \frac{\beta_t}{\sqrt{1-\bar{\alpha}_t}}\epsilon_\theta(\boldsymbol{z}_t, t)\right) \tag{20}$$

is obtained by a neural network that predicts the added noise using a neural network $\epsilon_\theta(\boldsymbol{z}_t, t)$.

A more suitable inference distribution for ending with a compact objective can be expressed via the reverse posterior conditioning on the observation $\boldsymbol{z}_0$:

$$q(\boldsymbol{z}_{t-1}|\boldsymbol{z}_t, \boldsymbol{z}_0) = \mathcal{N}\left(\boldsymbol{z}_{t-1}; \tilde{\boldsymbol{\mu}}_t(\boldsymbol{z}_t, \boldsymbol{z}_0), \tilde{\beta}_t\boldsymbol{I}\right), \tag{21}$$

The model is learned by minimizing the variational bound on negative log-likelihood, which can be expressed as a sum of terms,

$$\mathcal{L}_{\text{DDPM}} = \overbrace{D_{\text{KL}}\left(q(\boldsymbol{z}_T|\boldsymbol{z}_0)\,\|\,p(\boldsymbol{z}_T)\right)}^{const} + \overbrace{-\log p(\boldsymbol{z}_0|\boldsymbol{z}_1)}^{L_0} + \\ \underbrace{D_{\text{KL}}\left(q(\boldsymbol{z}_{t-1}|\boldsymbol{z}_t, \boldsymbol{z}_0)\,\|\,p_\theta(\boldsymbol{z}_{t-1}|\boldsymbol{z}_t)\right)}_{L_{t-1}} \tag{22}$$

Efficient training is achieved by uniformly sampling $t \sim U(1, T)$ and optimizing the corresponding $L_{t-1}$, which, by deriving Equation (22), can be further simplified to a denoising score-matching loss:

$$L_{\text{DDPM}} = \mathbb{E}_{\boldsymbol{z}_0, t, \epsilon}\left[\|\epsilon - \epsilon_\theta(\boldsymbol{z}_t, t)\|^2\right] \tag{23}$$

where $\epsilon \sim \mathcal{N}(0, \mathbf{I})$.

A.2.1. INFERENCE

The reverse posterior density defined in Equation (21) is no longer Markovian and coincides with the inference model proposed in DDIM (Song et al., 2021a), where it is demonstrated that faster sampling can be achieved without retraining, simply by redefining the generative process as:

$$p_\theta(\boldsymbol{z}_{t-1}|\boldsymbol{z}_t) = \begin{cases} \mathcal{N}\left(f_\theta^{(1)}(\boldsymbol{z}_1), \sigma_1^2 \boldsymbol{I}\right) & \text{if } t = 1 \\ q_\sigma\left(\boldsymbol{z}_{t-1}|\boldsymbol{z}_t, f_\theta^{(t)}(\boldsymbol{z}_t)\right) & \text{otherwise,} \end{cases} \tag{24}$$

and considering that an estimation of $\hat{z}_0 = f_\theta(\boldsymbol{z}_t, t)$ can be computed as:

$$f_\theta(\boldsymbol{z}_t, t) = \frac{1}{\sqrt{\bar{\alpha}_t}}\left(\boldsymbol{z}_t - \sqrt{1 - \bar{\alpha}_t} \cdot \epsilon_\theta(\boldsymbol{z}_t, t)\right), \tag{25}$$

where $\bar{\alpha}_t = \prod_{i=1}^t \alpha_i$. This formulation enables deterministic sampling with improved efficiency using fewer steps.

# B. Training `LDMI`

---

**Algorithm 1** Hyper-Transforming LDM

---

**Input:** Dataset $\{(\boldsymbol{X}_i, \boldsymbol{Y}_i)\}_{i=1}^N$, pre-trained LDM (frozen encoder $\mathcal{E}_\psi$, frozen diffusion model $p_\theta$), Hyper-Transformer decoder $p_\Phi$

---

**repeat**
    Sample batch $(\boldsymbol{X}, \boldsymbol{Y}) \sim p_{\text{data}}(\boldsymbol{X}, \boldsymbol{Y})$
    Sample latent using frozen encoder: $\boldsymbol{z} \sim q_\psi(\boldsymbol{z}|\boldsymbol{X}_m, \boldsymbol{Y}_m)$
    Compute likelihood loss: $\mathcal{L}_{\text{HT}}(\phi)$ using Equation (15)
    **if** image data **then**
        Add perceptual loss: $\mathcal{L}_{\text{percept}}$
        Add adversarial loss: $\mathcal{L}_{\text{adv}}$
    **end if**
    Update decoder parameters $\phi$ to minimize total loss
**until** convergence

---

---

**Algorithm 2** Training `LDMI`

---

**Input:** Dataset $(\boldsymbol{X}_i, \boldsymbol{Y}_i)i = 1^N$, encoder $\mathcal{E}_\psi$, decoder $p_\Phi$, diffusion model $p_\theta$

---

**Stage 1**: Training `I-VAE`
**repeat**
    Sample batch $(\boldsymbol{X}, \boldsymbol{Y}) \sim p_{\text{data}}(\boldsymbol{X}, \boldsymbol{Y})$
    Sample latent: $\boldsymbol{z} \sim q_\psi(\boldsymbol{z}|\boldsymbol{X}, \boldsymbol{Y})$
    Compute ELBO loss: $\mathcal{L}_{\text{VAE}}(\phi, \psi)$ using Equation (11)
    **if** image data **then**
        Add perceptual loss: $\mathcal{L}_{\text{percept}}$
        Add adversarial loss: $\mathcal{L}_{\text{adv}}$
    **end if**
    Update parameters $\phi, \psi$ to minimize total loss
**until** convergence

---

**Stage 2**: Training DDPM
**repeat**
    Sample batch $(\boldsymbol{X}, \boldsymbol{Y}) \sim p_{\text{data}}(\boldsymbol{X}, \boldsymbol{Y})$
    Sample latent: $\boldsymbol{z} \sim q_\psi(\boldsymbol{z}|\boldsymbol{X}, \boldsymbol{Y})$
    Sample noise: $\epsilon \sim \mathcal{N}(\boldsymbol{0}, \boldsymbol{I})$
    Sample timestep: $t \sim U(1, T)$
    Compute DDPM loss: $\mathcal{L}_{\text{DDPM}}$ using Equation (14)
    Update parameters $\theta$ to minimize $\mathcal{L}_{\text{DDPM}}$
**until** convergence

---

In addition to the hyper-transforming approach described in the main text, `LDMI` can also be trained from scratch using a two-stage process. This follows the standard Latent Diffusion Model (LDM) training pipeline but incorporates our Hyper-Transformer Decoder (`HD`) for INR generation. The training procedure consists of:

1. **Stage 1: Learning the Latent Space with I-VAE** — We train a Variational Autoencoder for INRs (`I-VAE`) to encode continuous signals into a structured latent space. The encoder learns an approximate posterior $q_\psi(z|X, Y)$, while the decoder reconstructs signals from the latent variables. The training objective follows the Evidence Lower Bound (ELBO) as defined in Equation (11). To enhance reconstruction quality, additional perceptual or adversarial losses may be applied for certain data types.

2. **Stage 2: Training the Latent Diffusion Model (LDM)** — Given the structured latent space obtained in Stage 1, we fit a diffusion-based generative model $p_\theta(z)$ to the aggregate posterior $q_\psi(z)$. This stage follows the standard DDPM framework, minimizing the objective in Equation (14). The learned diffusion prior enables generative sampling in the latent space, from which the `HD` decoder infers INR parameters.

The full training procedure is summarized in **Algorithm 2**.

## C. Additional Experiments

### C.1. Scalability of `LDMI`

A key strength of our approach lies in its parameter efficiency and scalability when compared to alternative methods. As shown in Table 3, while previous approaches such as GASP (Dupont et al., 2022b) or VAMoH (Koyuncu et al., 2023) require substantial hypernetwork parameters (25.7M) to generate relatively small INR weights (50K), `LDMI` achieves superior performance with only 8.06M hypernetwork parameters while generating 330K INR weights for a 5-layer network.

| Method | HN Params | INR Weights | Ratio (INR/HN) |
|---|---|---|---|
| GASP/VAMoH | 25.7M | 50K | 0.0019 |
| LDMI | **8.06M** | **330K** | **0.0409** |

Table 3: Parameter efficiency of hypernetworks (HN) in GASP/VaMoH and `LDMI`.

We confirm this efficiency effect through an ablation study comparing our transformer-based `HD` decoder against a standard MLP design, as MLPs are commonly used in hypernetwork implementations despite their limitations in modeling complex dependencies. Table 4 shows that on CelebA-HQ, our `HD` architecture not only achieves superior reconstruction quality (with a 2.79 dB improvement in PSNR) but does so with significantly fewer parameters—less than half compared to the MLP architecture.

| Method | HN Params | PSNR (dB) |
|---|---|---|
| LDMI-MLP | 17.53M | 24.93 |
| LDMI-HD | **8.06M** | **27.72** |

Table 4: Ablation study comparing MLP and hyper-transformer `HD` decoders on CelebA-HQ.

This parameter efficiency highlights a crucial advantage of our approach: hyper-transformer's ability to capture complex inter-dimensional dependencies translates to more effective weight generation with a more compact architecture. The superior scaling properties of `LDMI` suggest that it can handle larger and more complex INR architectures without prohibitive parameter growth, making it particularly suitable for high-resolution generation tasks that demand larger INRs.

### C.2. Samples at Multiple Resolutions

To further demonstrate the flexibility of our model, we present additional qualitative results showcasing unconditional samples generated at varying resolutions. As described in the main text, we sample from the latent diffusion prior, decode the latents into INRs using our `HD` decoder, and evaluate them on coordinate grids of increasing resolution.

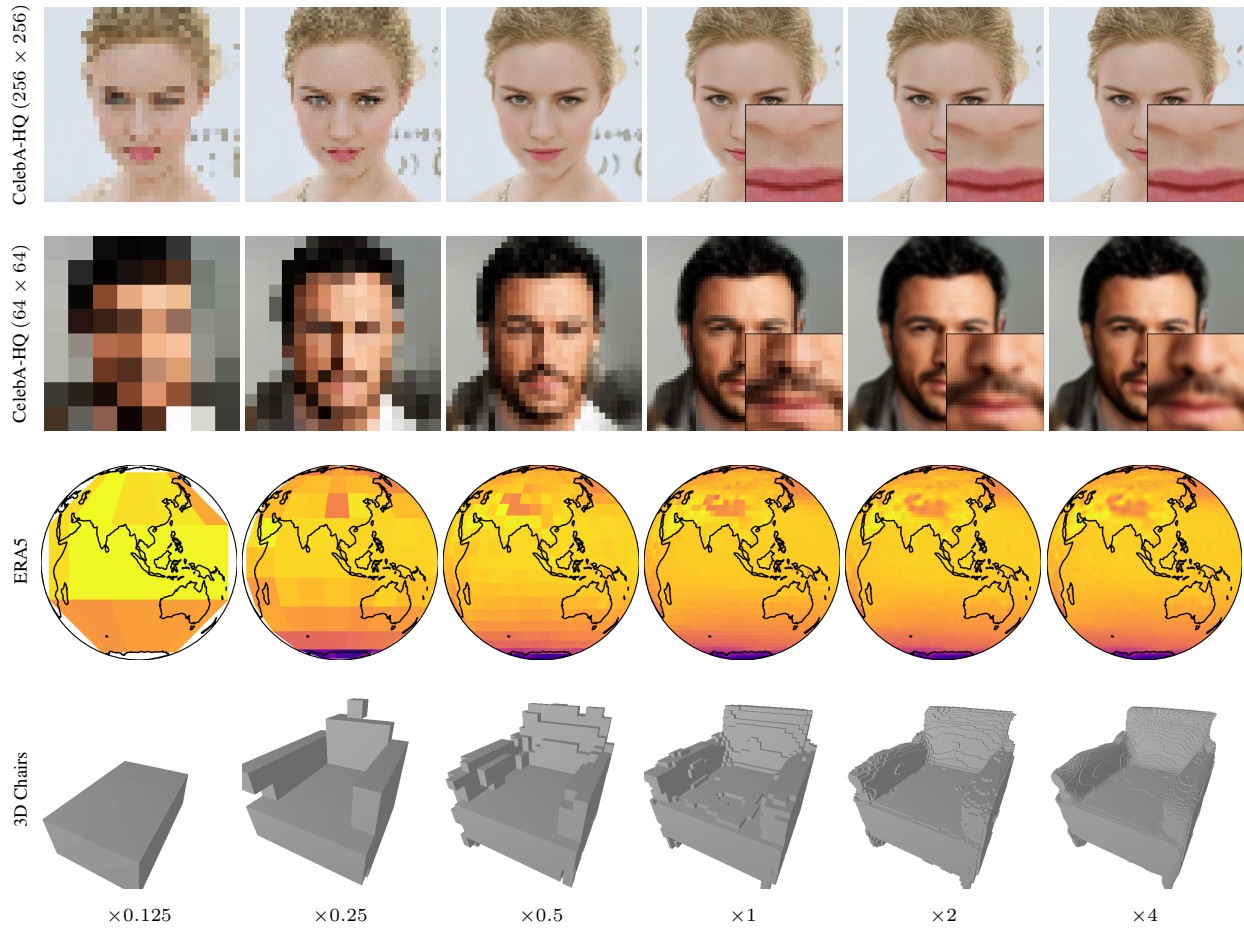

Figure 8: Additional uncurated samples from LDMI at multiple resolutions.

Figure 8 displays generations from LDMI trained on data across different modalities, rendered at scales ranging from ×0.125 to ×4. These results highlight the resolution-agnostic nature of our approach and its ability to produce coherent, high-quality outputs across a wide range of spatial resolutions.

### C.3. Reconstructions at Multiple Resolutions

We provide additional reconstruction results to further illustrate LDMI's ability to operate seamlessly across a wide range of output resolutions. Each input image is encoded into a latent representation, sampled from the posterior, and decoded into an INR using our HD decoder. The resulting INR is then evaluated over coordinate grids of increasing density to generate reconstructions at progressively higher resolutions.

Figures 9 and 10 present reconstructions from LDMI trained on CelebA-HQ at (256 × 256) and (64 × 64), respectively. Reconstructions are rendered at resolutions ranging from ×0.125 to ×4, showcasing the model's ability to maintain spatial coherence and fine details even under extreme upsampling. Notably, such flexible reconstruction is not supported by baseline methods like GASP due to architectural limitations. Furthermore, datasets of this complexity are not even addressed by other reconstruction-capable baselines such as VAMoH (Koyuncu et al., 2023), Functa (Dupont et al., 2022b), or Spatial Functa (Bauer et al., 2023).

## D. Experimental Details

This section outlines the hyperparameter settings used to train LDMI across datasets. We describe key components, including the encoder, decoder, diffusion model, and dataset settings. Table 5 summarizes the hyperparameters used in our experiments,

covering both stages of the generative framework: (i) the first-stage autoencoder—either a VQVAE or $\beta$-VAE, depending on the dataset—and (ii) the second-stage latent diffusion model. It lists architectural choices such as latent dimensionality, diffusion steps, attention resolutions, and optimization settings (e.g., batch size, learning rate), along with details of the `HD` decoder, tokenizer, Transformer modules, and INR architecture. For image-based tasks, we use perceptual and adversarial losses to improve sample quality, following Esser et al. (2021). All models were trained on `NVIDIA H100` GPUs.

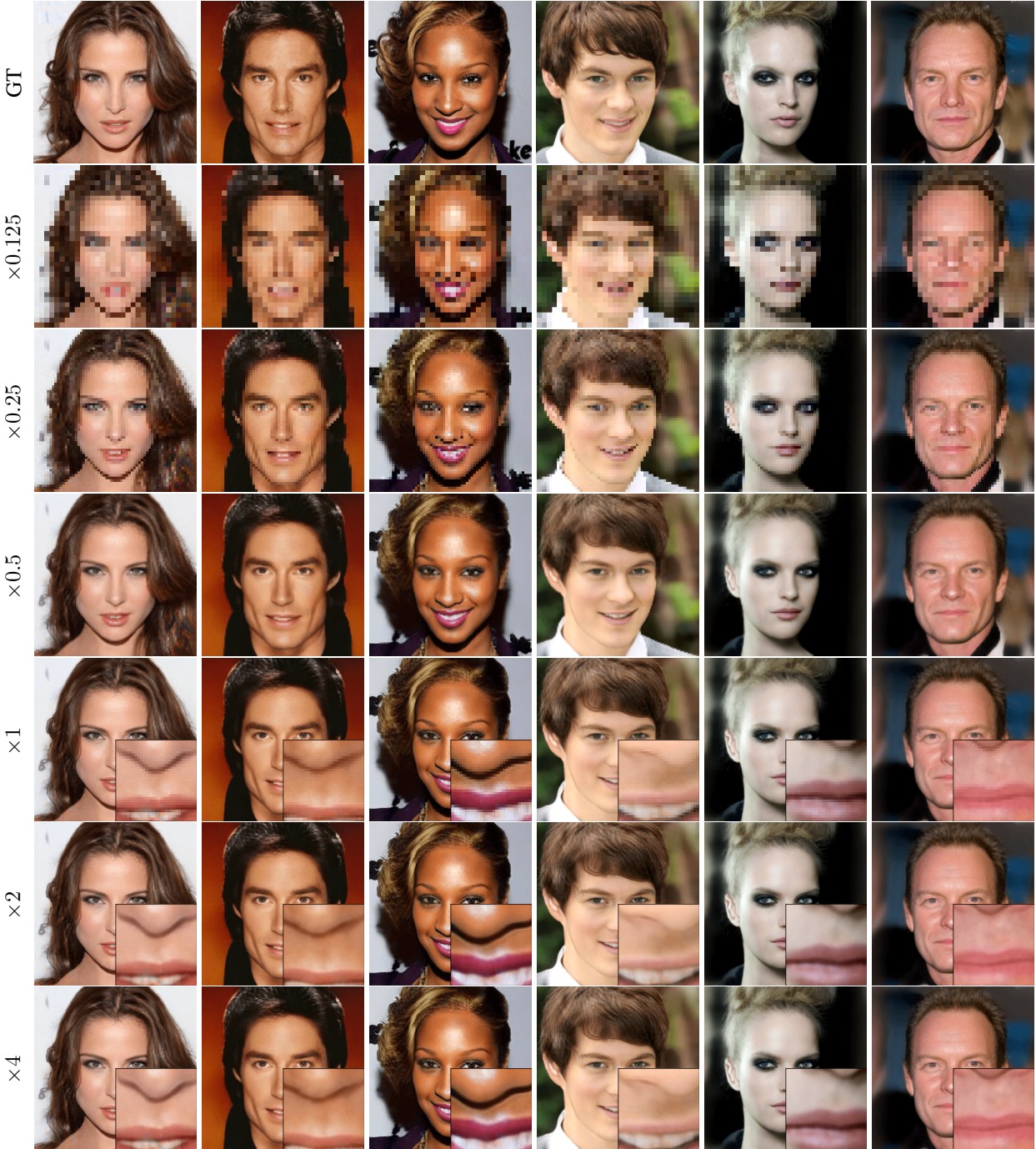

Figure 9: Reconstructions of test CelebA-HQ ($256 \times 256$) images by LDMI at multiple resolutions. The ground truth image is first passed through the encoder, which produces the parameters of the posterior distribution. A latent code is then sampled and transformed into the parameters of the INR using our HD decoder. By simply evaluating the INRs at denser coordinate grids, we can generate images at increasingly higher resolutions.

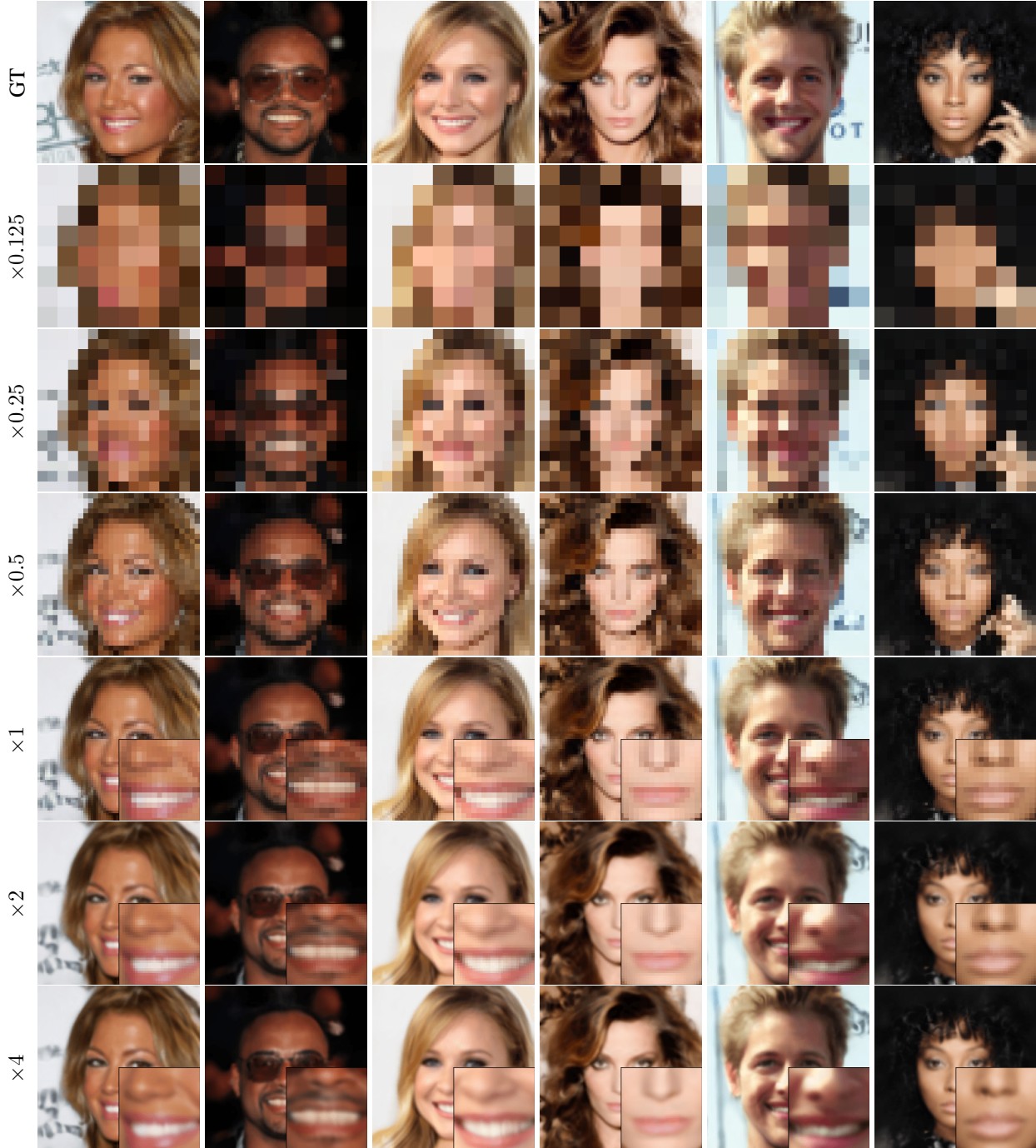

Figure 10: Reconstructions of test CelebA-HQ ($64 \times 64$) images by `LDMI` at multiple resolutions. The ground truth image is first passed through the encoder, which produces the parameters of the posterior distribution. A latent code is then sampled and transformed into the parameters of the INR using our `HD` decoder. By simply evaluating the INRs at denser coordinate grids, we can generate images at increasingly higher resolutions.

| | CelebA | CelebA-HQ $64 \times 64$ | CelebA-HQ $256 \times 256$ | ImageNet | ERA5 | Chairs |
|---|---|---|---|---|---|---|
| resolution | $64 \times 64$ | $64 \times 64$ | $256 \times 256$ | $256 \times 256$ | $46 \times 90$ | $32 \times 32 \times 32$ |
| modality | image | image | image | image | polar | occupancy |
| first_stage_model | $\beta$-VAE | $\beta$-VAE | VQVAE | VQVAE | $\beta$-VAE | $\beta$-VAE |
| **Encoder** | | | | | | |
| codebook_size | - | - | 8192 | 8192 | - | - |
| latent_channels | 3 | 3 | 3 | 3 | 3 | 64 |
| base_channels | 64 | 64 | 128 | 128 | 32 | 32 |
| ch_mult | 1,2,4 | 1,2,4 | 1,2,4 | 1,2,4 | 1,2,4 | 1,2,4 |
| num_blocks | 2 | 2 | 2 | 2 | 2 | - |
| dropout | - | 0.1 | - | - | - | 0.2 |
| kl_weight | 1e-05 | 1e-04 | - | - | 1.0e-6 | 1.0e-6 |
| perc_weight | 1. | 1. | 1. | 1. | - | - |
| **Discriminator** | | | | | | |
| layers | 2 | 2 | 3 | 3 | - | - |
| n_filters | 32 | 64 | 64 | 64 | - | - |
| dropout | - | 0.2 | - | - | - | - |
| disc_weight | 0.75 | 0.75 | 0.75 | 0.6 | - | - |
| **HD decoder** | | | | | | |
| **Tokenizer** | | | | | | |
| latent_size | $16 \times 16$ | $16 \times 16$ | $64 \times 64$ | $64 \times 64$ | $11 \times 22$ | $4 \times 4$ |
| patch_size | 2 | 2 | 4 | 4 | 1 | 1 |
| heads | 4 | 4 | 4 | 4 | 4 | 4 |
| head_dim | 32 | 32 | 32 | 32 | 32 | 32 |
| **Transformer** | | | | | | |
| token_dim | 192 | 192 | 384 | 768 | 136 | 128 |
| encoder_layers | 6 | 6 | 5 | 6 | 4 | 3 |
| decoder_layers | 6 | 6 | 5 | 6 | 4 | 3 |
| heads | 6 | 6 | 6 | 12 | 4 | 4 |
| head_dim | 48 | 48 | 64 | 64 | 32 | 32 |
| feedforward_dim | 768 | 768 | 1536 | 3072 | 512 | 512 |
| groups | 64 | 64 | 128 | 128 | 64 | 64 |
| dropout | - | 0.1 | 0.1 | - | - | 0.2 |
| **INR** | | | | | | |
| type | SIREN | SIREN | SIREN | MLP | SIREN | |
| layers | 5 | 5 | 5 | 5 | 5 | 5 |
| hidden_dim | 256 | 256 | 256 | 256 | 256 | 256 |
| point_enc_dim | - | - | - | - | 256 | - |
| $\omega$ | 30. | 30. | 30. | 30. | - | 30. |
| **Latent Diffusion** | | | | | | |
| shape_z | $3 \times 16 \times 16$ | $3 \times 16 \times 16$ | $3 \times 64 \times 64$ | $3 \times 64 \times 64$ | $3 \times 11 \times 22$ | $64 \times 4 \times 4$ |
| —z— | 768 | 768 | 12288 | 12288 | 726 | 1024 |
| diffusion_steps | 1000 | 1000 | 1000 | 1000 | 1000 | 1000 |
| noise_schedule | linear | linear | linear | linear | linear | linear |
| base_channels | 64 | 32 | 224 | 192 | 64 | 128 |
| ch_mult | 1,2,3,4 | 1,2,3,4 | 1,2,3,4 | 1,2,3,5 | 1,2,3,4 | 1,2 |
| attn_resolutions | 2, 4, 8 | 2, 4, 8 | 8, 16, 32 | 8, 16, 32 | - | 4 |
| head_channels | 32 | 32 | 32 | 192 | 32 | 64 |
| num_blocks | 2 | 2 | 2 | 2 | 2 | 2 |
| class_cond | - | - | - | crossattn | - | - |
| context_dim | - | - | - | 512 | - | - |
| transformers_depth | - | - | - | 1 | - | - |
| **Training** | | | | | | |
| batch_size | 64 | 64 | 32 | 32 | 64 | 128 |
| iterations | 300k, 400k | 300k, 200k | 1M | 1.4M | 400k, 500k | 300k, 60k |
| lr | 1e-06, 2e-06 | 1e-06, 2e-06 | 4.5e-6 | 1.0e-6 | 1e-06, 2e-06 | 1e-06, 2e-06 |
| hyper-transforming | - | - | ✓ | ✓ | - | - |
| LDM version | - | - | VQ-F4 (no-attn) | VQ-F4 (no-attn) | - | - |

Table 5: Architecture details.

