# OpenReview forum: "Hyper-Transforming Latent Diffusion Models"
_ICML.cc/2025/Conference — ICML 2025 poster_

### Official Review · Reviewer_XroW · 2025-03-13

**Overall Recommendation:** 4

**Summary:**

This work introduces a novel "LDMI" framework which empowers latent diffusion models to generate Implicit Neural Representations (INRs). The proposed  Hyper-Transformer Decoder enables the space of INR parameters to be learned in a flexible and probabilistic manner. Empirical tests are conducted on a range of image and shape reconstruction tasks.

**Claims And Evidence:**

Most claims within the paper are well supported by the experimental evidence. Perhaps the statement on "our work establishes [....] generative modelling with unconstrained resolution" is a bit lacking in evidence since output resolutions seem to match training resolutions in the experiments.

**Essential References Not Discussed:**

To the best of my knowledge, no essential references are missing.
Somewhat more tangentially, there are some complementary works on performing functional inference with diffusion models which may be of interest such as:
Neural Diffusion Processes, Dutordoir et al, ICML 2023
All-in-one simulation-based inference, Gloecker et al, ICML 2024

**Experimental Designs Or Analyses:**

In the CelebA experiment in FIgure 4, the baselines are trained on a different dataset to the LDMI. While the claim is made that this makes the task easier for the baselines, this lack of a controlled experiment prevents rigorous conclusions to be drawn from the differences in performance.

**Methods And Evaluation Criteria:**

A good variety of datasets are explored, and for the case of ImageNet and CelebA, standard metrics are appropriately used such as PSNR and FID. It would certainly be preferable to also see some quantitative metrics for the ERA5 and ShapeNet chairs however - few robust scientific conclusions can be drawn by eye.

**Other Comments Or Suggestions:**

I would recommend making it explicitly clear that pixel values correspond to integrals over a finite region, so are not ideally suited for this application, but they serve more as a useful testbed.

**Other Strengths And Weaknesses:**

It ought to be viable to perform inference at a higher resolution than what was used during training, as it's one of the key advantages of possessing a INR, but I don't see explicit examples, perhaps some examples could be added to the supplementary material?

**Questions For Authors:**

Relating to the above point, I feel this paper would be stronger if first it spends a bit of time clarifying the key problem/challenge they wish to tackle, before then outlining the solution.

**Relation To Broader Scientific Literature:**

This work sits in the popular field of diffusion models and hypernetworks, and provides a good review of relevant prior works in the area.

**Theoretical Claims:**

No, while some theoretical background is given, the key results here are empirical.

---

> ### Author Rebuttal · Authors · 2025-03-31
>
> We thank the reviewer for the constructive feedback and encouraging remarks. Your comments helped us significantly improve the manuscript. Below, we address all the concerns raised.
>
> ## On Our Claims Regarding Unconstrained Resolution
> We agree that validating our model’s ability to generalize to unseen coordinates is essential. Following your suggestion, we added new experiments on super-resolution. During [reconstruction](https://anonymous.4open.science/api/repo/LDMI_pre-7F42/file/experiments/figures/super_recs_celebahq256.png?v=5cc02656), test images at the training resolution are encoded, decoded into INRs, and evaluated on denser grids. For [sampling](https://anonymous.4open.science/api/repo/LDMI_pre-7F42/file/experiments/figures/super_samples.png?v=485b880b), latent codes are drawn from the diffusion prior and decoded similarly. These experiments confirm that $\texttt{LDMI}$ captures continuous signals and generalizes to higher resolutions—highlighting a key advantage of INR-based generation.
>
>
> ## New quantitative results for ERA5 and ShapeNet
> To better support our claims of modality generalization, we now report the PSNR measured on the ERA5 and Chairs datasets:
>
> | Method    | Chairs   | ERA5    |
> | --------  | -------- |-------- |
> | Functa     |    29.2     |    34.9     |
> | VAMoH     |    38.4      |    39.0     |
> | $\texttt{LDMI}$     |     **38.8**      |     **44.6**     |
>
> We omit GASP (not applicable to reconstruction due to their GAN-based framework). $\texttt{LDMI}$ generalizes without adaptation and outperforms comparable models.
>
> ## On CelebA and CelebA-HQ
> You are correct—the original submission trained on CelebA, while baselines used CelebA-HQ. We now explicitly distinguish them and retrain $\texttt{LDMI}$ on CelebA-HQ at $64 \times 64$ for a fair comparison. To further demonstrate scalability, we train $\texttt{LDMI}$ on CelebA-HQ at $256\times 256$, achieving strong results not addressed by prior work.
>
>
> ## On related work: NDPs and SimFormer
> Thank you for pointing out these complementary works. We agree that approaches such as Neural Diffusion Processes, and SimFormer, are interesting and relevant within the broader context of function-space modeling with diffusion processes. In the following, we highlight the main difference.
>
> **NDPs** leverage a diffusion model to define distributions over function values at given coordinates. Their architecture explicitly enforces exchangeability and permutation invariance via a bi-dimensional attention mechanism, and their sampling mechanism mimics Gaussian processes and related meta-learning methods such as Neural Processes. **Importantly**, the function itself is not represented via a neural network whose parameters are generated or learned—rather, the model learns to denoise function values directly, conditioned on inputs.
>
> **Simformer** is designed for simulation-based inference (SBI), where the goal is to infer unknown parameters of stochastic simulators from observations. It treats both data and parameters as random variables and learns a diffusion model over the joint distribution $p(\boldsymbol{x}, \boldsymbol{\theta})$, allowing for flexible sampling of any conditional (e.g., posterior, likelihood, marginals). Parameters may include function-valued (infinite-dimensional) components, but they are not represented as INRs—rather, they are input variables within the inference pipeline. Simformer excels at amortized Bayesian inference with unstructured or missing data and flexible conditioning.
>
> We added these references and discussion to the Related Work section of our revised manuscript.
>
> ## On the interpretation of pixel values as integrals
> We agree that pixels approximate integrals over regions. Following [SIREN, NeRF, Functa], we adopt the standard Dirac delta approximation by treating pixel values as samples at center coordinates. We’ve made this assumption explicit in the revised paper.
>
>
> ## On the core challenge we address
> Thank you for raising this point. We have clarified the motivation in the revised manuscript.
>
> Our goal is to overcome the scalability bottlenecks of MLP-based hypernetworks in generative modeling of functions via INRs. While INRs excel at representing continuous signals, prior methods (e.g., Functa, GASP, VAMoH) require hypernetworks with tens of millions of parameters—even for small 3-layer INRs—making scaling impractical.
>
> To address this, we propose the $\texttt{HD}$ decoder, a Transformer-based hypernetwork that maps latent samples to INR weights. Its parameter count remains fixed as INR size grows, with only sequence length increasing. We further optimize efficiency with a grouping strategy, enabling generation of large INRs at low cost.
>
> Integrating this design into latent diffusion models allows $\texttt{LDMI}$ to model more complex signals and generalize across resolutions efficiently.

---

> > ### Comment · Reviewer_XroW · 2025-04-08
> >
> > I appreciate the thoughtful response, and have updated my score accordingly.

---

> > > ### Author Response · Authors · 2025-04-08
> > >
> > > Dear Reviewer XroW,
> > >
> > > Thank you very much for your thoughtful engagement during the rebuttal phase and for updating your score. We truly appreciate the time and care you took to evaluate our responses. Your feedback helped us improve the paper, and we're very happy to hear that it is now in a form you consider ready for acceptance.
> > >
> > > Best regards,
> > >
> > > The authors.

---

### Official Review · Reviewer_WKA5 · 2025-03-13

**Overall Recommendation:** 3

**Summary:**

This paper proposes a new framework for INR generation (LDMI) which combines latent diffusion models and a transformer based hyper network for learning the distributions over INR parameters. The hyper network transforms the latent variables through a transformer encoder and decoder and generates the INR parameters.
Beyond the vanilla end-to-end training pipeline of a LDMI, it enables also using a pre-trained LDM and a hyper transformer for efficient transfer learning without full retraining.

**Claims And Evidence:**

The authors claim that their method is effective in generation tasks as well as hyper-transforming tasks. However, the results from Table 1 show that on CelebA LDMI doesn't achieve comparable PSNR score compared to Functa and the FID score is much higher than GASP. Also on ImageNet the PSNR is not comparable with Spatial Functa although the FID score does outperform its rival.

**Essential References Not Discussed:**

Ruiz, Nataniel, et al. "Hyperdreambooth: Hypernetworks for fast personalization of text-to-image models." Proceedings of the IEEE/CVF conference on computer vision and pattern recognition. 2024.

The above mentioned paper uses a hyper network approach to personalize text-to-image models which should be relevant to this paper.

**Experimental Designs Or Analyses:**

They show qualitative results of their method on generation, reconstruction, hyper-transforming and data completion tasks. However, the experimental results are not sufficient to show the effectiveness of their approach.

**Methods And Evaluation Criteria:**

The datasets they evaluate on are reasonable, including multiple standard image datasets.
The metrics they use are standard. (PSNR, FID)
However, they do not provide evidence that the model can achieve comparable performance with other baselines with either less parameters or less training samples etc.

**Other Comments Or Suggestions:**

No

**Other Strengths And Weaknesses:**

Strength:
The idea is well explained.
Weakness:
The quantitive results of the paper are not strong enough to show the effectiveness of the proposed approach.

**Questions For Authors:**

In the hyper-training setting, which frozen LDM do you use, I don't think it's specified in the paper.
You also mentioned that it is only been trained for a limited number of iterations and achieve the qualitative results shown in Figure 3b, do you have results regarding metrics vs training time to show the effectiveness of your approach?

**Relation To Broader Scientific Literature:**

This paper is connected to latent diffusion models, transformer-based hypernetworks also the implicit neural representation (INR) literature.

**Theoretical Claims:**

I have not spotted error in the proofs of any theoretical claims.

---

> ### Author Rebuttal · Authors · 2025-03-31
>
> We thank the reviewer for the constructive feedback and for highlighting relevant connections to related work. Below, we address each of the raised concerns in detail.
>
> ## On the relation to HyperDreamBooth
> We appreciate your suggestion to consider HyperDreamBooth, which we have now cited and discussed in the revised manuscript. While the goals of our work differ, there are indeed interesting architectural parallels. Both approaches leverage transformer-based hypernetworks to generate weights, motivated by scalability and efficiency.
>
> HyperDreamBooth focuses on personalizing pre-trained text-to-image diffusion models by generating low-rank residuals (via LoRA) from a single image—enabling rapid adaptation to new subjects. In contrast, $\texttt{LDMI}$ introduces a generative model over the space of implicit neural representations (INRs), which represent continuous functions across diverse modalities (e.g., images, 3D occupancy fields, and climate data).
>
> Rather than modulating a pre-trained model, our Hyper-Transformer Decoder generates the full parameter set of an INR from latent samples, enabling resolution-agnostic, function-level generation. While both methods rely on Transformer-based hypernetworks, $\texttt{LDMI}$ operates in a distinct regime of generative modeling over functions.
>
> ## On the competitiveness of our method and strengthened evaluation
> Thank you for raising this point. In response, we have significantly strengthened our empirical results to more clearly demonstrate the advantages of our approach.
>
> First, we added results on higher-complexity datasets such as CelebA-HQ $(256 \times 256)$, which fall outside the scope of prior baselines. Our model demonstrates strong performance in both [generation](https://anonymous.4open.science/api/repo/LDMI_pre-7F42/file/experiments/figures/super_samples.png?v=485b880b) and [reconstruction](https://anonymous.4open.science/api/repo/LDMI_pre-7F42/file/experiments/figures/super_recs_celebahq256.png?v=5cc02656) at multiple resolutions—without retraining—showcasing the benefits of INR-based generation.
>
> Second, we updated Table 1 with new results and included the number of hypernetwork parameters to highlight $\texttt{LDMI}$’s key strength: scalability. For example:
> - GASP/VAMoH require 25.7M parameters to generate ~50K INR weights.
> - $\texttt{LDMI}$ uses 8.06M parameters to generate ~330K INR weights for a deeper, 5-layer INR.
>
> Despite having significantly fewer parameters, $\texttt{LDMI}$ delivers superior performance.  For details, please refer to our [response to Reviewer ER44](https://openreview.net/forum?id=yhgcRwJ9Dn&noteId=ldvx3foEnj), where the updated table is provided.
>
> To further contextualize this comparison, we highlight some key aspects of the baselines:
> - Functa relies on **test-time optimization**, fitting each test INR with access to ground-truth data. While this explains its high PSNR, it departs from our amortized inference setting and limits fair comparison. We include it for completeness given the task similarity.
> - GASP **cannot perform reconstructions** due to its GAN-based design, limiting its use in conditional tasks.
>
> In summary, $\texttt{LDMI}$ offers a compelling trade-off between quality, scalability, and generalization, supported by stronger experiments in the revised version.
>
> ## On the hyper-training setting and training efficiency
>
> Thank you for this question. In our hyper-transforming setup, we use the publicly available pre-trained LDM from [Rombach et al., 2022], specifically the LDM-VQ-4 variant, trained on CelebA-HQ at $(256 \times 256)$ resolution, and the LDM-VQ-8, trained on ImageNet. We freeze both the encoder and the diffusion-based prior, replacing only the decoder with our $\texttt{HD}$ module. While Section 3.5 of the original submission specified the frozen components, we have now revised the text to make this distinction more explicit.
>
> This setup provides a key advantage: significantly faster training. By freezing most of the architecture and training only the decoder, we reduce the number of learnable parameters and avoid the need for a full two-stage training pipeline—common in diffusion models—where the autoencoder and the prior must be learned sequentially. Our hyper-transforming approach thus offers a lightweight, modular alternative that efficiently adapts pre-trained models to new functional decoding regimes.

---

### Official Review · Reviewer_LB9r · 2025-03-17

**Overall Recommendation:** 3

**Summary:**

The authors propose a novel method for generating the parameters of implicit neural representations (INRs) representing real data. They use a latent diffusion framework, which first trains a VAE to learn a rich latent representation of data, then trains a diffusion generative model on the learned representations to generate new samples. The main innovation is the introduction of a hyper-transformer decoder, which uses an encoder-decoder transformer network to translate from encoded latent representations to INR parameters. Besides training the VAE and diffusion from scratch, the authors also introduce a hyper-transforming paradigm, in which a pre-trained encoder and diffusion model are frozen and only the hyper-decoder weights are updated. Experiments demonstrate that the proposed method performs comparably to baselines.

#### Updates after rebuttal period
The authors have made sincere efforts to address the points I raised, and the new changes and additions to the manuscript strengthen the writing and clarify the mathematical details. In light of these revisions, I will raise my original review score.

**Claims And Evidence:**

The claims are backed up.

**Essential References Not Discussed:**

N/A.

**Experimental Designs Or Analyses:**

Experiments appear sound.

**Methods And Evaluation Criteria:**

The methods appear sound.

**Other Comments Or Suggestions:**

- Line 238, right column: "$d_{i}n$" should be "$d_{in}$"

**Other Strengths And Weaknesses:**

## Strengths

- The paper is well-written. There is a clear flow of ideas from background to motivation which sets the stage for the explanation of the main innovations and the results. In addition, the authors take care to explain most of their design choices and the mathematical objects that comprise these designs to clarify their purpose (with a few mistakes and/or errors, as explained in the weaknesses section).

- The hyper-transformer decoder is a well-motivated way to convert latent representations to INR parameters. Specifically, the use of learnable template weights and biases is a clever way to reduce computational and memory complexity.

- The hyper-transforming training paradigm allows the re-use of powerful, existing models trained on large datasets. This can drastically reduce the computational requirements of the method compared to training from scratch.


## Weaknesses

- Since this is a novel idea, the authors should make the notation as easily understandable as possible - especially for the hyper decoder (HD). However, the repeated usage of, e.g., $\mathbf{W}$ for weight matrices with only subscripts and superscripts to differentiate between the types of weights makes understanding the HD more difficult. This is not a large weakness, as I believe the authors present their method quite clearly otherwise, but may be something to consider for future revisions.

- Similar to the point above, the dimensions of the weights are inconsistent. For example, in line 264 the authors state that the dimensions of $\bar{W}^i$ are $d_{out} \times G$ and Figure 2 shows each $\bar{W}^i$ having sequence length $G$, implying that sequence length corresponds to the number of columns. However, in line 272, $\bar{W}^b$ are stated to have dimensions $d_{in} \times d_{out}$ but the sequence length given in Figure 2 is $d_{in}$ which corresponds to the number of rows instead of the columns. This makes it difficult to reconcile the interactions between the weights and hidden states.

- The right hand side of Eq (7) is a (normalized) dot product of two vectors, which should result in a scalar value and not in a column vector as desired. If the authors mean to use the dot operator ($\cdot$) as an element-wise product between the two vectors, they should clarify this to avoid confusion.

- The results seem to show that the proposed method is not much better, if at all, than competing baselines. Table 1 indicates that other methods achieve either superior reconstruction accruracy or superior generative ability. The ImageNet samples in Figure 3b appear unnatural and unconverged.

**Questions For Authors:**

- How did you choose the specific normalization of the output weight matrix as shown in Eq (7)? I.e., why did you choose to normalize the output by the L2 norm of the product of the two input columns?

- In the hyper-transforming training setup, how do you pass the coordinate inputs to the encoder? Pre-trained VAE models used in latent diffusion generally do not take this information as part of their inputs.

**Relation To Broader Scientific Literature:**

The field of diffusion models, and LDMs in particular, has been growing as the de facto most popular method of generative modeling. While normally restricted to fixed grids of pixels, the introduction of implicit neural representations to this modeling paradigm represents an important shift toward a more flexible representation of the inputs and outputs.

**Theoretical Claims:**

There are no theoretical claims.

---

> ### Author Rebuttal · Authors · 2025-03-31
>
> We thank the reviewer for the thoughtful and constructive feedback, and for recognizing the clarity, motivation, and contributions of our work. Your comments helped improve the paper significantly.  Below, we address each of the concerns raised in your review.
>
> ## On the notation used for weights
> We appreciate your comments on the novelty of our weight generation strategy and agree that clearer notation helps with readability. In the revised version, we have clarified the notation as follows:
>
> - The bar in $\bar{\mathbf{W}}$ denotes globally shared, learnable parameters (e.g., $\bar{\mathbf{W}}^\text{i}$ and $\bar{\mathbf{W}}^\text{b}$).
> - The superscript $i$ in $\bar{\mathbf{W}}^\text{i}$ refers to “*input*” to the Transformer Decoder. The superscript $\text{o}$ in $\mathbf{W}^\text{o}$ refers to its “*output*”, which transforms these global input weights by attending to the latent tokens.
> - The subscript $l$ in $\mathbf{W}_l$ identifies the target INR layer. We omit this when not needed for clarity.
> - The superscript $\text{b}$ in $\bar{\mathbf{W}}^\text{b}$ stands for the “*base*” weights, which serve as the template modulated by $\mathbf{W}^\text{o}$ via our grouping and reconstruction strategy.
>
> These clarifications are now reflected in the manuscript and Figure 2.
>
> ## On the correctness Equation 7 and the weight dimensions
> We highly appreciate that you pointed this out, you are completely right. We have corrected the following two issues:
> - We corrected the typo in line 272 to have $\bar{\mathbf{W}}^\text{b} \in \mathbb{R}^{d_{\text{out}}\times d_{\text{in}}}$.
> - Now we use the Hadamard or element-wise product $\odot$ for the weight reconstruction.
>
> In Equation (7), each $w_{\lfloor c / k\rfloor}^{\mathrm{o}}$ (we're skipping bold notation here due to rendering issues) is an output token of the Transformer, and also the $\lfloor c / k\rfloor$-th column of the grouped weight matrix $\mathbf{W}^{\text{o}} \in \mathbb{R}^{d_{\text{out}}\times G}$. The length of the Transformer decoder sequences are thus $L\times G$, which is the total number of grouped columns. We can set the embedding dimension of the Transformer to $d_{out}$ or project the tokens using one feed-forward layer, depending on the case.
>
> Additionally, we corrected the typo in line 238 to have $d_\text{in}$. We hope after these corrections, the concern is fully addressed.
>
> ## On the competitiveness of our method
> Our key strength is scalability: we outperform baselines while using fewer hypernetwork parameters. For instance:
> - GASP/VAMoH: 25.7M params → 50K INR weights
> - $\texttt{LDMI}$: 8.06M params → 330K INR weights (5-layer network)
>
> We also extended Table 1 and added experiments on resolution generalization at [generation](https://anonymous.4open.science/api/repo/LDMI_pre-7F42/file/experiments/figures/super_samples.png?v=485b880b) and reconstruction, including [CelebA-HQ $(256 \times 256)$](https://anonymous.4open.science/api/repo/LDMI_pre-7F42/file/experiments/figures/super_recs_celebahq256.png?v=5cc02656), which are not tackled by prior work. We refer to our [response to Reviewer ER44](https://openreview.net/forum?id=yhgcRwJ9Dn&noteId=ldvx3foEnj), where we include the updated Table. To clarify the context behind other baselines:
> - Functa uses test-time optimization, tuning each test sample with ground-truth.
> - GASP cannot perform reconstructions due to its GAN-based design.
>
> In sum, $\texttt{LDMI}$ offers a strong balance of quality, scalability, and generalization.
>
> ## On the choice of the weight reconstruction method
> Thank you for this insightful question. Initially, we adopted a normalization-based reconstruction (inspired by Trans-INR), which worked well for MLP-based INRs. However, this approach led to instability when generating SIREN weights, particularly due to vanishing gradients—hindering generalization in tasks like super-resolution.
>
> To address this, we introduced a novel scaling-based reconstruction:
> $$
> \left(1+w_{\lfloor c / k\rfloor}^{\text{o}}\right) \odot \bar{w}_c^{\text{b}}.
> $$
>
> This avoids collapse when $\boldsymbol{w}_{\lfloor c / k\rfloor}^{\mathrm{o}} \approx 0$ and provides stable training for high-frequency signals—leading to the strong results in our revision.
>
> ## On coordinate input handling in the hyper-transforming setup
> Thank you for raising this question. Our goal is to model data as samples from a stochastic process, using our $\texttt{HD}$ decoder to map latents into functions. In the hyper-transforming setup, we use convolutional encoders (often pre-trained) that implicitly capture coordinate information from structured grids:
> - For image and ERA5 data: ResNet encoders
> - For 3D occupancy fields: 3D convolutional encoders
>
> The decoder then produces INR weights, enabling evaluation at arbitrary coordinates and decoupling resolution from representation.

---

> > ### Comment · Reviewer_LB9r · 2025-04-04
> >
> > Many thanks to the authors for their thorough rebuttal and for addressing my questions. I will keep my review and my score the same as before, as I still believe that the paper needs more re-writing before it is ready for publication.

---

> > > ### Author Response · Authors · 2025-04-04
> > >
> > > Dear Reviewer LB9r,
> > >
> > > Thank you again for your thoughtful review and for acknowledging the clarity and contributions of our work. However, we were disappointed to see your score remain unchanged, especially given that your final comment (*“the paper needs more re-writing”*) introduced a concern not mentioned in your original review.
> > >
> > > This new claim also stands in contrast with your initial assessment, in which you wrote:
> > >
> > > > *“The paper is well-written. There is a clear flow of ideas from background to motivation which sets the stage for the explanation of the main innovations and the results.”*
> > >
> > > Throughout the rebuttal, we made substantial efforts to address every point you raised:
> > > - We clarified Equation (7), corrected all identified typos, and clearly explained the use of element-wise operations in the reconstruction.
> > > - We revised the manuscript to improve the clarity and consistency of the weight notation, as you suggested.
> > > - We strengthened our empirical results with new datasets, higher-resolution experiments, and clearer comparisons—demonstrating the scalability and competitiveness of our approach.
> > > - We explained our change from normalization to scaling in the weight reconstruction, which significantly improved model stability with SIREN-based INRs.
> > > - We responded in detail to your question on coordinate inputs in the hyper-transforming setup, clarifying the role of convolutional encoders.
> > >
> > > We took your feedback seriously and used it to significantly improve the manuscript. Considering the scope of the improvements made in direct response to your comments, we were surprised that neither feedback nor an updated evaluation was provided. If there are remaining issues beyond those already raised and addressed, we would sincerely welcome the opportunity to respond to them.
> > >
> > > Best regards,
> > >
> > > The authors.

---

### Official Review · Reviewer_ER44 · 2025-03-24

**Overall Recommendation:** 2

**Summary:**

This paper introduces a new generative framework called Latent Diffusion Models of Implicit Neural Representations (LDMI), which integrates Implicit Neural Representations (INRs) into transformer-based latent diffusion models. The key component is to use a Hyper-Transformer Decoder (HD) to replace traditional MLP-based hypernetworks and address their limitations of scalability and efficiency. This module generates INR parameters from latent variables. Experiments have shown the effectiveness of the proposed framework.

**Claims And Evidence:**

About the claim of scalability of hyper-transformers. The current version of this paper lacks clear experimental evidence to show that there is scalability in hyper-transformers.

**Essential References Not Discussed:**

The paper lacks a discussion on the following works about diffusion transformers:

[A] Bao F, Nie S, Xue K, et al. All are worth words: A vit backbone for diffusion models[C]//Proceedings of the IEEE/CVF conference on computer vision and pattern recognition. 2023: 22669-22679.

[B] Peebles W, Xie S. Scalable diffusion models with transformers[C]//Proceedings of the IEEE/CVF international conference on computer vision. 2023: 4195-4205.

**Experimental Designs Or Analyses:**

Yes, I checked the experimental design. A key problem with this paper is the lack of comparison to the advanced diffusion models [A, B, C] or a baseline of diffusion models with MLP-based hyper-networks. This comparison is essential to the claim of scalability of hyper-transformers.

[A] Dhariwal P, Nichol A. Diffusion models beat gans on image synthesis[J]. Advances in neural information processing systems, 2021, 34: 8780-8794.

[B] Bao F, Nie S, Xue K, et al. All are worth words: A vit backbone for diffusion models[C]//Proceedings of the IEEE/CVF conference on computer vision and pattern recognition. 2023: 22669-22679.

[C] Peebles W, Xie S. Scalable diffusion models with transformers[C]//Proceedings of the IEEE/CVF international conference on computer vision. 2023: 4195-4205.

**Methods And Evaluation Criteria:**

Yes. The methods and evaluation criteria make sense.

**Other Comments Or Suggestions:**

Please refer to my questions in the “weakness” section.

**Other Strengths And Weaknesses:**

Strengths:

- The motivation of using hyper transformers to address the scalability limitation of traditional hypernets is clear.
- This paper is well-written and easy to follow.

Weaknesses:
- The claim of scalability has not been verified. In line 60, the authors claim that the proposed hyper-transformer decoder can solve the scalability of hyper-networks. However, the authors only verify that the hyper-transformer can work on ImageNet in Table 1. There is no evidence of whether the larger hyper-transformer achieves better performance than the MLP-based hyper-nets of a similar size.
- Table 1 only compares the method, Spatial Functa. The authors have not compared their method to the advanced diffusion models [A, B, C] or a baseline of diffusion models with MLP-based hyper-networks.

[A] Dhariwal P, Nichol A. Diffusion models beat gans on image synthesis[J]. Advances in neural information processing systems, 2021, 34: 8780-8794.

[B] Bao F, Nie S, Xue K, et al. All are worth words: A vit backbone for diffusion models[C]//Proceedings of the IEEE/CVF conference on computer vision and pattern recognition. 2023: 22669-22679.

[C] Peebles W, Xie S. Scalable diffusion models with transformers[C]//Proceedings of the IEEE/CVF international conference on computer vision. 2023: 4195-4205.

**Questions For Authors:**

Please refer to my questions in the “weakness” section.

**Relation To Broader Scientific Literature:**

This paper is a new method of introducing the hyper-nets into diffusion models for image generation and upgrading the MLP-based hyper-nets into hyper-transformers.

**Theoretical Claims:**

Yes. I have checked the theory of diffusion models and hyper-nets discussed in this paper.

---

> ### Author Rebuttal · Authors · 2025-03-31
>
> We thank the reviewer for the detailed and thoughtful feedback, as well as for recognizing the novelty and clarity of our work. Below, we address the key concerns raised.
>
> ## On the scope and nature of the contribution
> $\texttt{LDMI}$ is not intended to compete with standard diffusion models that operate in pixel space. These models approximate $p(\boldsymbol{y})$ over discrete grids. In contrast, we model $p(\boldsymbol{y}|\boldsymbol{x})$ as a stochastic process, where $\boldsymbol{y}$ is a function represented via INR parameters $\theta$. This defines a substantially richer and more complex generative space—orthogonal to that of the suggested references. To our knowledge, $\texttt{LDMI}$ is the first to use Transformers as decoders to generate INR weights from latent samples in this setting.
>
> That said, we appreciate the reviewer’s suggestion and agree that including and discussing these works helps contextualize our contributions. We have added the suggested pixel-based diffusion models [A, B, C] to the Related Work section and explicitly discussed how our problem setting fundamentally differs.
>
> ## On baseline comparisons
> To provide fair comparisons in the INR setting, we benchmark against the closest functional baselines. In addition, to emphasize the differences with respect to the provided references, we have added super-resolution experiments (e.g., [samples](https://anonymous.4open.science/api/repo/LDMI_pre-7F42/file/experiments/figures/super_samples.png?v=485b880b) and [reconstructions](https://anonymous.4open.science/api/repo/LDMI_pre-7F42/file/experiments/figures/super_recs_celebahq256.png?v=5cc0265)) to show that LDMI generalizes across resolutions without retraining—a key property enabled by its function-based design.
>
> ## On scalability of the $\texttt{HD}$ Decoder
> While our original submission already compared $\texttt{LDMI}$ against MLP-based hypernetworks (used in all baselines), we have now further strengthened our scalability analysis. The updated Table 1 includes hypernetwork sizes:
>
> | **Model**                         | **PSNR (dB) ↑** | **FID ↓** | **HN Params ↓** |
> |----------------------------------|-----------------|-----------|-----------------|
> | **CelebA-HQ (64 × 64)**          |                 |           |                 |
> | GASP       | N/A               | **7.42**  | 25.7M           |
> | Functa     | $\mathbf{\leq 30.7}$      | 40.40     | N/A               |
> | VAMoH      | 23.17           | 66.27     | 25.7M           |
> | **$\texttt{LDMI}$**                         | 27.72           | 11.08      | **8.06M**       |
> |                                  |                 |           |                 |
> | **ImageNet (256 × 256)**         |                 |           |                 |
> | Spatial Functa  | $  \mathbf{\leq 38.4} $   | $ \geq 8.5  $| N/A               |
> | **$\texttt{LDMI}$**                         | 20.69           | **6.94**  |  102.78M
>
> GASP and VAMoH use ~25.7M parameters to generate 50K weights for shallow 3-layer INRs. In contrast, our $\texttt{HD}$ decoder uses **only 8.06M parameters** to generate 330K weights for a deeper 5-layer INR. Despite using 70% fewer parameters, $\texttt{LDMI}$ performs better, scaling to complex signals like [CelebA-HQ $(256 \times 256)$](https://anonymous.4open.science/api/repo/LDMI_pre-7F42/file/experiments/figures/super_recs_celebahq256.png?v=5cc0265)—not tackled by any baseline.
>
> To show superiority against latent diffusion and MLPs, as suggested, the table below compares $\texttt{LDMI}$ using either an MLP or our $\texttt{HD}$ on CelebA-HQ:
>
> | Method                          | HN Params               | PSNR (dB)  |
> | ------------------------------- | ----------------------- | ---------- |
> | $\texttt{LDMI}$-MLP             | 17.53                   | 24.93      |
> | $\texttt{LDMI}$-$\texttt{HD}$   | **8.06M**               | **27.72**  |
>
>
> ## On the competitiveness of our method
> We believe a few clarifications are helpful for contextualizing the results in Table 1.
>
> Functa’s high PSNR arises from **test-time optimization**: it fits a separate modulation vector per test image using ground truth, unlike our amortized inference approach. This undermines a fair comparison, but we include it for completeness and clarify this in the revised manuscript.
>
> While GASP reports lower FID on CelebA, it **cannot perform reconstructions** due to its GAN-based design, limiting its use in conditional tasks.
>
> Considering these factors, LDMI provides a balanced solution—competitive in both sampling and reconstruction—while being more scalable and efficient.
>
> ## Discussion on Diffusion Transformers
> We appreciate the suggested references and have added them to the Related Work section. These approaches apply Transformers as denoising backbones in pixel-space diffusion. Our use is fundamentally different: we apply Transformers as hypernetworks to generate INR weights, enabling function-level generation beyond the limitations of discrete grids.

---

### Decision · Program_Chairs · 2025-05-01

**Decision:**

Accept (poster)

**Comment:**

This paper proposes the use of Implicit Neural Representations within a Transformer-based hypernetwork, framework for improving the scalability of the latent variable models. The key idea is to replace decoders with transformer hypernetworks.

The paper received four reviews with mixed opinions. Issues were raised regarding the validity of the key claim of scalability vis a vis the competing method (GASP), notational inconsistency and writing.

While writing can be improved in the revision, I do believe that the following comment by the Reviewer ER44, on paper's key claim on scalability warrants a deeper examination -

"This suggests that GASP leverages its larger parameter count to achieve superior performance, showing the scaling ability of GASP rather than LDMI. This experiment fails to demonstrate that LDMI can similarly enhance performance by scaling parameters or weights"

I read through the paper and liked the idea  and encourage the Authors to take a relook at that comment and thoroughly back their key claims with more experiments.